# Johnson-Lindenstrauss Lemma Guided Network for Efficient 3D Medical Segmentation

**Jinpeng Lu**[1,*], **Linghan Cai**[2,*], **Yinda Chen**[1,*], **Guo Tang**[3], **Songhan Jiang**[3],
**Haoyuan Shi**[1], **Zhiwei Xiong**[1,†]

[1] University of Science and Technology of China
[2] Dresden University of Technology
[3] Harbin Institute of Technology, Shenzhen
`jinplu@mail.ustc.edu.cn, zwxiong@ustc.edu.cn`

## Abstract

Lightweight 3D medical image segmentation remains constrained by a fundamental *"efficiency / robustness conflict"*, particularly when processing complex anatomical structures and heterogeneous modalities. In this paper, we study how to redesign the framework based on the characteristics of high-dimensional 3D images, and explore data synergy to overcome the fragile representation of lightweight methods. Our approach, VeloxSeg, begins with a deployable and extensible dual-stream CNN-Transformer architecture composed of Paired Window Attention (PWA) and Johnson-Lindenstrauss lemma-guided convolution (JLC). For each 3D image, we invoke a "glance-and-focus" principle, where PWA rapidly retrieves multi-scale information, and JLC ensures robust local feature extraction with minimal parameters, significantly enhancing the model's ability to operate with low computational budget. Followed by an extension of the dual-stream architecture that incorporates modal interaction into the multi-scale image-retrieval process, VeloxSeg efficiently models heterogeneous modalities. Finally, Spatially Decoupled Knowledge Transfer (SDKT) via Gram matrices injects the texture prior extracted by a self-supervised network into the segmentation network, yielding stronger representations than baselines at no extra inference cost. Experimental results on multimodal benchmarks show that VeloxSeg achieves a 26% Dice improvement, alongside increasing GPU throughput by $11\times$, CPU by $48\times$, and reducing training peak GPU memory usage by $1/20$, inference by $1/24$. Code is available at `https://github.com/JinPLu/VeloxSeg`.

## 1 Introduction

3D medical image segmentation serves as a cornerstone of contemporary clinical workflows (Wu et al., 2025; Peiris et al., 2023; Wang et al., 2026), driving rapid advances in semantic segmentation models (Liu et al., 2024a; Shaker et al., 2024; He et al., 2025; Yu et al., 2025a; Wald et al., 2025; Cai et al., 2023; 2024). However, translating these advances into clinical practice faces significant obstacles, including limited hardware resources, stringent latency requirements, and the need to achieve multi-organ generalization while handling heterogeneous multimodal data in deployment environments. These challenges have spurred the development of lightweight 3D medical segmentation methods, leading to lightweight approaches with fewer than 5 million parameters (Perera et al., 2024; Pang et al., 2024; Yu et al., 2025b; Li et al., 2025; Ye et al., 2025). Yet, the pursuit of smaller parameter counts and lower computational costs has revealed a fundamental and increasingly prominent trade-off: these lightweight models struggle to maintain both efficiency and robust performance when handling heterogeneous data and complex lesions, which we term *"efficiency / robustness conflict"*. We address this problem from two key perspectives:

**Insufficient consideration of the high-dimensional complexity of 3D data.** Recent sequence models, such as Mamba (Gu & Dao, 2023; Xing et al., 2025) and RWKV (Peng et al., 2023; Ye

---

*Equal contribution.
†Corresponding author.

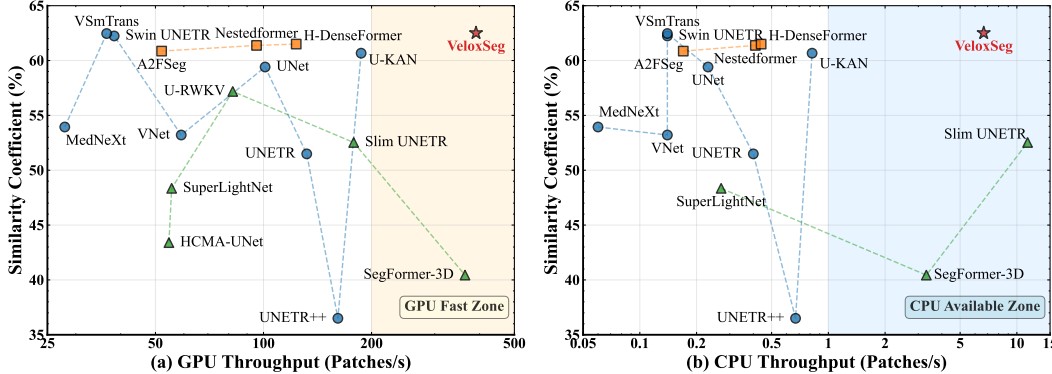

Figure 1: Comparison of our proposed VeloxSeg with recent methods on the AutoPET-II test set. Basic models, multimodal models, lightweight models, and our model are marked with circles, squares, triangles, and stars, respectively. GPU and CPU Throughput are measured on an NVIDIA RTX3090 GPU and a single-core Intel(R) Xeon(R) Gold 5320 CPU, respectively.

et al., 2025), have achieved remarkable progress in segmentation, owing to their linear complexity and long-range modeling capabilities. However, due to the lack of more efficient scanning strategies suitable for 3D data, these methods have not yet supplanted CNN-Transformer architectures in the domain of efficient medical segmentation. Our model is built on a dual-stream CNN-Transformer architecture, synergizing the complementary strengths of both components: the inductive bias (Le-Cun et al., 1989; Mansour et al., 2019) and training stability (He et al., 2016; Ioffe & Szegedy, 2015; Ulyanov et al., 2016) of convolutions with the global modeling power (Vaswani et al., 2017) and extensibility (Lu et al., 2019; Chen et al., 2020; Dosovitskiy et al., 2020) of Transformers. Pruning is the most common approach to model lightweighting (Molchanov et al., 2019; Fang et al., 2023), but its final configuration relies on dataset-specific importance metrics and hand-tuned sparsity schedules, which limit generalizability and lead to expensive retraining. Therefore, developing lightweight and efficient components is essential. Constructing relationships among tokens is the core of feature modeling. In principle, self-attention (Dosovitskiy et al., 2021) can represent arbitrary dependencies, but in practice it is constrained by computation and memory. Window-based attention (Hatamizadeh et al., 2021; Du et al., 2025) performs fine-grained relation modeling within local windows, but it relies on cascaded operations to capture cross-window interactions, leading to substantial redundancy. Axial attention (Liu et al., 2024a) and downsampled attention (Pang et al., 2024; Perera et al., 2024; Kuang et al., 2025) accelerate the construction of relationships between a token and distant tokens by constraining attention paths or operating at lower resolutions, but they tend to weaken the representation of critical local dependencies. We propose paired window attention (PWA), which builds parallel multi-scale feature streams and coordinates short- and long-range attention to capture global token relations while maintaining sufficient focus on local information, at a computational cost comparable to axial or downsampled attention. Convolution with its inductive bias remains indispensable for detailed local modeling. However, common depthwise-separable designs (Chollet, 2017; Ma et al., 2018; Roy et al., 2023; Muhammad et al., 2025) suffer from a key limitation: aggressive channel decoupling disrupts the original geometric adjacency among tokens, making them harder to distinguish and fragmenting the information. This issue is particularly severe for complex anatomical structures and heterogeneous modalities. To address this, we introduce a Johnson–Lindenstrauss (JL) lemma–guided lightweight convolution (Lindenstrauss & Johnson, 1984), which enforces a minimum number of channels per group in each convolution layer to preserve geometric adjacency among tokens. This design keeps the model lightweight while ensuring that fine-grained details can be robustly captured.

**Insufficient exploration of data synergy, including multimodal cooperation and data priors.** Exploiting multimodal complementary information is crucial for robust model representation (Mu et al., 2020; Zheng et al., 2025; Zou et al., 2025). However, it is often ignored by lightweight methods due to the potential increase in computational cost, even when training on multimodal datasets. As discussed in Appendix B, bridging multimodal information across multiple scales is vital for extracting complementary information from heterogeneous modalities. Therefore, we extend our dual-stream architecture, using PWA to facilitate efficient modal interaction at the additional cost

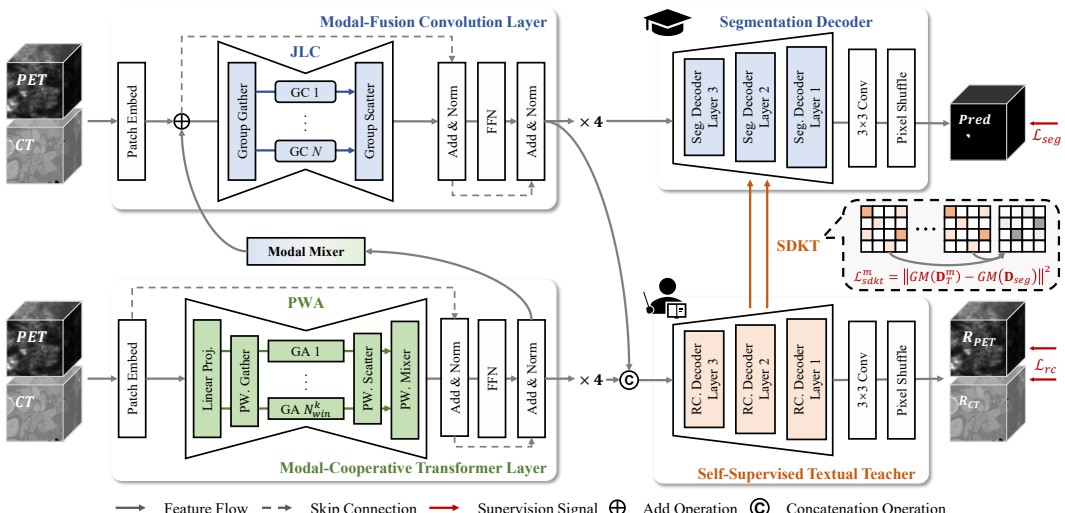

Figure 2: Overview of VeloxSeg. VeloxSeg employs an encoder-decoder architecture with Paired Window Attention (PWA) and Johnson-Lindenstrauss lemma-guided convolution (JLC) on the left, using 1×1 convolution as modal mixer. GC: group convolution; GA: multimodal grouped attention.

of only 0.27 MParams and 0.09 GFLOPs. Besides, exploring prior knowledge from existing data to enhance a model's detailed representation holds practical significance for efficient segmentation methods. These methods often achieve higher efficiency by performing segmentation in a compressed space, which comes at the cost of exploring small lesions and complex boundaries (Perera et al., 2024; He et al., 2023; Rahman et al., 2024; Pang et al., 2024). Although establishing cross-task knowledge transfer from reconstruction to segmentation appears to be a solution (Sun et al., 2020; Rui et al., 2025; Zhang et al., 2020; Wang et al., 2025b), the significant differences in their regions of interest (ROIs) often lead to negative knowledge transfer (Qiu et al., 2023). To this end, our proposed Spatially Decoupled Knowledge Transfer (SDKT) is a simple yet effective solution, motivated by the observation that a common upsampling operation in reconstruction and super-resolution tasks, "Conv+PixelShuffle (Su et al., 2025)", essentially unfolds the channel relationships at each voxel position into the spatial details of the surrounding image patch. This suggests that the guidance provided by a texture teacher to a segmentation task should be based on the channel relationships within its features. The Gram matrix, commonly used to represent style in the field of image style transfer, characterizes feature channel relationships in a spatially-invariant manner (Gatys et al., 2016; 2015). Based on it, we establish a positive knowledge transfer path from a self-supervised texture teacher to the segmentation network with no inference overhead (Zhu et al., 2021; Akiva et al., 2022).

Inspired by the above insights, we propose VeloxSeg that systematically alleviates the "*efficiency / robustness conflict*" during model lightweighting. Extensive experiments thoroughly explored the rationale for the design choices and demonstrated the model's excellent clinical applicability and generalization capabilities. Figure 1 shows a comparison of VeloxSeg's performance with other methods on the AutoPET-II (Gatidis S, 2022) dataset, demonstrating strong competitiveness. In summary, we develop:

- A Paired Window Attention to ensemble multi-scale attention groups, capturing local-global information simultaneously, improving localization capabilities with less cost, and achieving low-cost but effective modal interaction at multiple scales.

- A Johnson-Lindenstrauss lemma-guided convolution that theoretically determines a minimum group size to preserve spatial adjacency, ensuring robust local feature extraction without costly and data-specific pruning.

- A Spatially Decoupled Knowledge Transfer strategy that uses Gram matrices to distill rich textural details from a self-supervised teacher during training, enhancing model fidelity with zero inference overhead.

## 2 METHODOLOGY

### 2.1 OVERVIEW OF VELOXSEG

As shown in Figure 2, VeloxSeg employs two 4-stage encoders, a modal-fusion convolution encoder and a modal-cooperative Transformer encoder, along with a segmentation decoder and a detail texture teacher. The Paired Window Attention (PWA), a key component of the transformer encoder, is designed to capture multi-scale and cross-modal context with low enough cost. The Johnson-Lindenstrauss lemma-guided Convolution (JLC), a key component of the convolution encoder, consists of 3 parallel JLCs at different scales to fuse modal information and model local features. Separating these two avoids a parameter explosion as the number of modalities increases, while maximizing the advantages and parallelism of both. In training, the Spatially Decoupled Knowledge Transfer (SDKT)

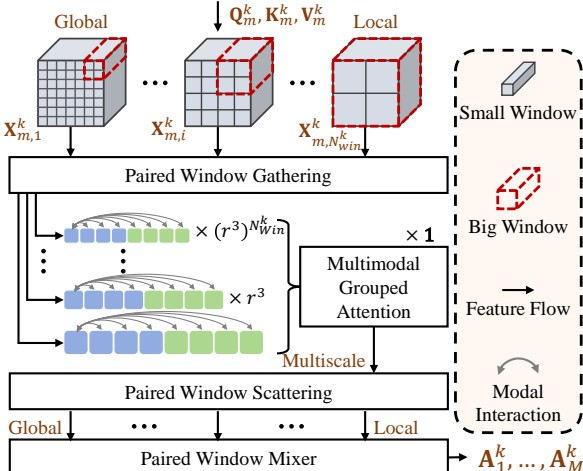

Figure 3: Overview of Paired Window Attention (PWA).

strategy is used to enhance texture representation, which is also of great significance for super-resolution and segmentation tasks.

### 2.2 PAIRED WINDOW ATTENTION

To achieve sufficiently strong clue-capturing capabilities with minimal computational cost, PWA ensembles parallel feature streams to capture key multimodal information at multiple scales. Notably, our approach differs significantly from conventional parallel multi-attention approaches (Liu et al., 2024a; Shaker et al., 2024), aiming to create a faster, lower-cost, more effective, and more elegant feature stream. Given $M$ modal features from $k$-th stage $\mathbf{E}_m^k, m = 1, \cdots, M$, they are first projected into $\mathbf{Q}_m^k, \mathbf{K}_m^k, \mathbf{V}_m^k$. As shown in Figure 3, we (i) partition the features into a set of big windows, collecting a salient token for each small window; (ii) synchronously expand window pairs to obtain multimodal sequences $\mathbf{X}_{m,i}^k, \mathbf{X} \in \{\mathbf{Q}, \mathbf{K}, \mathbf{V}\}$ of different scales but equal length, where $i$ is the number of the paired window; (iii) gather all sequences and compute attention across all scales and modalities at once; and (iv) use a lightweight mixer to simply and efficiently blend features from all scales. The attention $\mathbf{A}_m^k$ is obtained by the following formula:

$$\mathbf{A}_m^k = \text{PWA}\left(\mathbf{E}_m^k \mid \mathbf{E}_1^k, \cdots, \mathbf{E}_M^k\right). \tag{1}$$

For more information about PWA, please see the Appendix C, including PyTorch code, detailed formula flow, and complexity analysis. We also provide a detailed analysis of the necessity of multi-scale modeling of medical modalities. Notably, PWA requires only $\log(size), size \in \{H, W, D\}$ paired windows to capture global context, while the minimum window ensures the preservation of local details. PWA achieves near-linear complexity, with a linear coefficient of approximately 7.87% of Swin Transformer (Liu et al., 2021).

### 2.3 JOHNSON-LINDENSTRAUSS LEMMA-GUIDED CONVOLUTION

**Lemma 1** (Johnson-Lindenstrauss). *For any finite set $\mathcal{X} \subset \mathbb{R}^d$ with $|\mathcal{X}| = N$ and $\varepsilon \in (0, 1)$, there exists a linear map $f : \mathbb{R}^d \to \mathbb{R}^{d'}$ with $d' \geq c_{JL}\varepsilon^{-2}\log N$, all $x, y \in \mathcal{X}$ satisfy $(1 - \varepsilon)\|x - y\|_2 \leq \|f(x) - f(y)\|_2 \leq (1 + \varepsilon)\|x - y\|_2$.*

As shown in Figure 4, depth-wise convolution destroys the adjacency relationship between data in the feature space, making it difficult to connect the current clues with the key information of the case. Inspired by Likhosherstov et al. (2021) derivation of the minimum attention head size

via the Johnson-Lindenstrauss (JL) lemma, we build our lightest but robust convolution upon the above theory framework. In particular, we extend it to the 3D segmentation, exploring the lower bound on group size while preserving spatial adjacency. The JL lemma states that for $N$ points in high-dimensional space, we need at least $\mathcal{O}(\log N)$ embedding dimensions to preserve pairwise distances. The volume ratio of the input image of $M$ modalities to the intermediate feature is $v$, and each voxel of the feature must retain information from at least $v$ input voxels. Due to anatomical constraints and the boundedness of the normalized input values, the manifold $\mathcal{M}$ of segmentation-related information of the input image

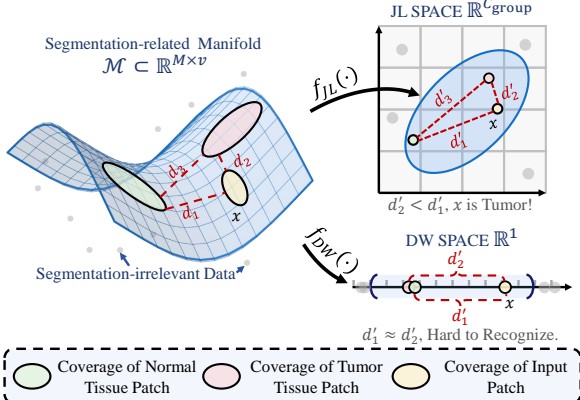

Figure 4: Intuitive difference between depth-wise (DW) convolution and Johnson-Lindenstrauss (JL) guided Convolution in the feature space.

patch can be covered by a finite number of samples, with a coverage count of $N(M, v)$. Substituting $N = N(M, v)$ into the lemma yields the size of the convolution group:

$$C_{\text{group}} = d' \geq c_{\text{JL}} \varepsilon^{-2} \log N(M, v), \tag{2}$$

where $C_{\text{group}}$ is the number of channels per group.

Due to the lack of $N$ in the vision domain, we empirically approximate $N(M, v)$ using $\hat{N}(M, v) = (M \cdot v)^{\alpha}$, where $\alpha$ is related to the difficulty of the segmentation task at hand. We conduct ablation studies on datasets with the richest modality heterogeneity and data distribution to identify the most generalizable scaling factor, which we use to obtain a lower bound on the group size of the convolution layers in each network stage. As analyzed in the Appendix D, we will use $\left\{C_{\text{group}}^k\right\}_{k=1}^4 = \{n, 2n, 2n, 4n\}$ as the group size for each stage of our network, where $n \in \mathbb{N}$ is determined from the most challenging AutoPET-II (Gatidis S, 2022) dataset to ensure multi-organ generalization capability. Appendix D further provides detailed derivations, including the covering-number motivated parametrization and the rationale behind our empirical approximation.

## 2.4 Spatially Decoupled Knowledge Transfer

To strengthen the representation of the lightweight model, we transfer the rich texture details extracted by the self-supervised texture teacher to the segmentation network via the Gram matrix. Specifically, we start with learning $M$ self-supervised detail texture teachers $T_m, m = 1, \cdots, M$, who are optimized by $M$ reconstruction tasks. The Gram matrix is commonly used to represent image style and can capture feature channel relationships in a spatially invariant manner. For feature maps $\mathbf{X} \in \mathbb{R}^{C \times (HWD)}$ with $C$ channels, the Gram matrix is:

$$\text{GM}(\mathbf{X}) = \frac{1}{CHWD} \left( \mathbf{X}\mathbf{X}^T \right) \in \mathbb{R}^{C \times C}. \tag{3}$$

SDKT is implemented by matching Gram matrices, which is mathematically equivalent to minimizing the maximum mean difference (MMD) using a second-order polynomial kernel (Li et al., 2017; Gupta et al., 2017). This naturally avoids a series of issues caused by excessive ROI discrepancies between the reconstruction/super-resolution features and the segmentation features. Specifically, a Gram-based consistency constraint serves as a positive knowledge transfer path between the segmentation features $\mathbf{D}_{seg}$ and the $M$ teacher features $\mathbf{D}_T^m$. Final loss $\mathcal{L}$ is:

$$\mathcal{L} = (\mathcal{L}_{dice} + \mathcal{L}_{ce}) + \lambda_{rc}\mathcal{L}_{rc} + \lambda_{sdkt} \sum_{m=1}^{M} \|\text{GM}(\mathbf{D}_T^m) - \text{GM}(\mathbf{D}_{seg})\|^2, \tag{4}$$

For information about $\mathcal{L}$, please see Appendix E. $\lambda_{seg}$, $\lambda_{rc}$ and $\lambda_{sdkt}$ are the loss weights.

## 3 EXPERIMENTS

### 3.1 DATASETS & METRICS

We validate the effectiveness of VeloxSeg on four public datasets: AutoPET-II (Gatidis S, 2022), Hecktor2022 (Oreiller et al., 2022), BraTS2021 (Baid et al., 2021), and BraTS2016 (Menze et al., 2014a) (details in Appendix F). Unlike typical medical segmentation datasets, the modality heterogeneity of PET/CT and the complex anatomical structures of multiple organs, and even the whole body, pose unique challenges to all models. We adopt comprehensive evaluation metrics suitable for clinical settings: Model Size (MParams), Computational Complexity (GFLOPs), Efficiency (GPU/CPU Throughput), and Segmentation Performance measured by Dice similarity coefficient (Dice) as the primary indicator, alongside 95% Hausdorff distance (HD95), Precision, and Recall.

### 3.2 IMPLEMENTATION DETAILS & BASELINES

Our implementation is based on PyTorch 2.4.1. Training is performed on an NVIDIA GeForce RTX 3090 GPU, while inference is run on an Intel(R) Xeon(R) Gold 5320 CPU. All datasets are standardized and partitioned into training, validation, and testing subsets in a 6:2:2 ratio. For training, we use a batch size of 4 with a 1:1 positive-to-negative sample ratio. Data augmentation involves random z-axis flipping with a 0.5 probability. We train the model for 300 epochs using the AdamW optimizer (Loshchilov & Hutter, 2017) with an initial learning rate of 2.5e-4 and a weight decay of 0.01. The learning rate is managed by a linear warmup and cosine annealing scheduler (Liu, 2022).

To ensure a convincing evaluation, we benchmark our method against a diverse set of models, including 8 basic models, 3 multimodal models, and 5 lightweight models, which are categorized accordingly in Tables 6. Furthermore, our analysis covers five distinct architectural paradigms: CNN-based models (UNet (Çiçek et al., 2016), VNet (Milletari et al., 2016), MedNeXt (Roy et al., 2023), A2FSeg (Wang & Hong, 2023)); CNN-Transformer hybrids (UNETR (Hatamizadeh et al., 2022), Nestedformer (Xing et al., 2022), SuperLightNet (Yu et al., 2025b)); CNN-KAN hybrids (U-KAN (Liu et al., 2024b)); CNN-Mamba hybrids (HCMA-UNet (Li et al., 2025)); and CNN-RWKV hybrids (U-RWKV (Ye et al., 2025)). The comparison is extended to include 2 advanced vision foundation models: SAM-Med3D (Wang et al., 2025a), which is evaluated in a zero-shot setting, and DINOv3 (Siméoni et al., 2025), for which the linear head is fine-tuned (Liu et al., 2025). Our comparison conforms to the fair comparison principle outlined in Isensee et al. (2024).

### 3.3 CLINICAL CAPABILITIES EVALUATION

Figure 1 provides a more intuitive comparison of the trade-offs between Dice and parameter count, and between Dice and GPU throughput. Specifically, regarding segmentation performance, Table 1 shows the segmentation performance for PET/CT. Appendix J shows the qualitative results of all models. Detailed computational costs are provided in the Appendix G. Furthermore, we report the GPU memory usage of all models on the three datasets, including training and inference, in Appendix H. To release the model's potential, we train VeloxSeg on the nnUNet (Isensee et al., 2021; Huang et al., 2023) training framework and compare it with the nnUNet baseline, as shown in Appendix I. In addition, to verify the modality adaptation ability of the method, we test the performance of MRI segmentation on BraTS2021.

**Comparison with Basic Models.** Against established basic architectures, including CNN-based, CNN-Transformer-based, and CNN-KAN-based methods, VeloxSeg demonstrates superior performance, with significantly lower computational cost. On the AutoPET-II dataset, VeloxSeg achieves a 62.51% Dice. This result marginally outperforms the best basic model, VSmTrans, using only 13.30% of its parameters and 1.96% of its GFLOPs. On Hecktor2022, VeloxSeg still surpasses all other models. These demonstrate that VeloxSeg is an efficient model in medical segmentation.

**Comparison with Multimodal Models.** When compared to specialized multimodal architectures, VeloxSeg demonstrates its effectiveness and efficiency in cross-modal feature integration. On the AutoPET-II dataset, VeloxSeg's Dice of 62.51% outperformed H-DenseFormer, Nestedformer, and A2FSeg by 1.01%, 1.13%, and 1.65%, respectively, while achieving GPU throughput improvements of 2.80× to 7.75× and a significant reduction of computational complexity. Furthermore,

| Method | Venue | AutoPET-II | | | | Hecktor2022 | | | |
|---|---|---|---|---|---|---|---|---|---|
| | | Dice ↑ | HD95 ↓ | Prec. ↑ | Rec. ↑ | Dice ↑ | HD95 ↓ | Prec. ↑ | Rec. ↑ |
| UNet | MICCAI'16 | 59.41 | 241.31 | 62.32 | 70.74 | 50.25 | 65.03 | 72.13 | 41.50 |
| VNet | 3DV'15 | 53.21 | 242.78 | 53.21 | 60.85 | 55.61 | 41.46 | 78.21 | 46.01 |
| MedNeXt-S | MICCAI'23 | 53.94 | 180.83 | 60.63 | 60.25 | 47.22 | 79.82 | 64.89 | 40.38 |
| UNETR | WACV'22 | 51.49 | 257.30 | 51.49 | 61.03 | 48.10 | 73.27 | 70.71 | 39.11 |
| Swin UNETR | MICCAI'21 | 62.24 | 242.07 | 62.91 | 73.30 | 44.56 | 103.02 | 62.43 | 37.55 |
| VSmTrans | MIA'24 | 62.46 | 223.88 | 65.19 | 70.92 | 52.91 | 78.03 | 61.91 | 50.97 |
| UNETR++ | TMI'24 | 36.50 | 178.57 | 36.50 | 60.16 | 29.95 | 27.74 | 61.84 | 21.75 |
| U-KAN | AAAI'25 | 60.67 | 70.91 | 62.03 | 72.94 | 55.89 | 23.48 | 77.72 | 46.89 |
| Nestedformer | MICCAI'22 | 61.38 | 265.51 | 61.38 | 64.29 | 40.17 | 72.95 | 63.22 | 32.59 |
| A2FSeg | MICCAI'23 | 60.86 | 131.48 | 60.86 | 76.10 | 40.90 | 32.95 | 77.02 | 30.57 |
| H-DenseFormer | MICCAI'23 | 61.50 | 252.98 | 61.41 | 75.76 | 46.79 | 34.84 | 78.33 | 35.31 |
| SAM-Med3D (CT) | TNNLS'25 | 13.13 | 101.24 | 19.82 | 16.70 | 27.52 | 18.84 | 43.94 | 24.46 |
| SAM-Med3D (PET) | TNNLS'25 | 26.59 | 101.94 | 31.92 | 31.86 | 31.94 | 18.03 | 69.69 | 24.35 |
| DINOv3-L (PET) | Arxiv'25 | 10.87 | – | 6.85 | 64.96 | 9.43 | – | 9.40 | 25.93 |
| DINOv3-L (CT+PET) | Arxiv'25 | 12.17 | – | 7.50 | 71.16 | 30.86 | – | 34.99 | 39.98 |
| SegFormer-3D | CVPRW'24 | 40.44 | 174.43 | 56.73 | 38.19 | 48.47 | 54.29 | 73.63 | 38.35 |
| Slim UNETR | TMI'24 | 52.53 | 310.53 | 53.99 | 66.55 | 49.40 | 56.55 | 69.53 | 41.20 |
| SuperLightNet | CVPR'25 | 48.35 | 59.09 | 60.82 | 47.61 | 50.03 | 34.36 | 75.29 | 40.65 |
| HCMA-UNet | ICME'25 | 43.40 | 146.11 | 43.32 | 62.46 | 42.06 | 146.11 | 67.68 | 33.18 |
| U-RWKV | MICCAI'25 | 57.18 | 61.12 | 66.69 | 59.40 | 45.97 | 56.83 | 64.52 | 39.71 |
| VeloxSeg | Ours | 62.51 | 241.08 | 67.76 | 66.28 | 56.48 | 47.66 | 74.81 | 49.24 |

i) Due to the small object and camouflage recognition involved, DINOv3-L (CT) cannot recognize tumors.
ii) "−" means that the value is out of range.

Table 1: Comparisons of segmentation performance on PET/CT datasets. The best performance is highlighted by red, followed by blue. VeloxSeg is highlighted in green.

on Hecktor2022, due to reduced data size, other multimodal models exhibit overfitting and overly conservative predictions, while VeloxSeg's Dice score remains stable.

**Comparison with Lightweight Models.** Against other lightweight methods, VeloxSeg is clearly superior. It leads in Dice on both datasets by a significant margin of over 5%. While some competitors have fewer parameters, they are computationally expensive or lack CPU support for clinical use. VeloxSeg offers the best balance, achieving 1.66 MParams and 1.79 GFLOPs. It also achieves a high GPU throughput of 599.06 patches/s and supports CPU-only devices, making it the most clinically practical solution.

**Comparison of Peak GPU Memory Usage.** As shown in Appendix H, VeloxSeg achieves the lowest or second lowest peak GPU memory usage among all methods. Compared to basic CNN/CNN-Transformer baseline models, VeloxSeg reduces memory usage by up to $20\times$ during training and up to $24\times$ during inference. Even among lightweight models, VeloxSeg is close to the most compact model (Slim UNETR), reducing memory usage by up to $10\times$ compared to other lightweight models.

**Train on nnUNet Training Framework.** As analyzed in Appendix I, VeloxSeg achieves a 14.2% Dice improvement with only 1.87% of nnUNet's MParams and 0.058% of its GFLOPs (Isensee et al., 2021), accompanied by a $4.8\times$ improvement in GPU throughput and a $52.5\times$ on CPU.

**Modality Adaptation Evaluation.** On the BraTS2021 dataset, which contains 4 MRI modalities, our early fusion strategy VeloxSeg-C achieves superior performance, surpassing the second-best method by 1.72% Dice. This demonstrates that our VeloxSeg can adapt to diverse multimodal segmentation tasks. Details can be found in Appendix K.

| Modules | | | Ablation | Hyper-Parameters | Params (M) | FLOPs (G) | Thr. GPU (Pat./s) | Dice (%) |
|---|---|---|---|---|---|---|---|---|
| Conv. | Trans. | SDKT. | | | | | | |
| ✓ | × | × | Width | $\langle 32, 64, 128, 256 \rangle$ | 2.65 | 5.31 | 145.63 | 48.96 |
| ✓ | × | × | | $\langle 16, 32, 64, 128 \rangle$ | **0.73** | 2.41 | 616.53 | 50.10 |
| ✓ | × | × | Kernel Size | $\langle 7 \rangle$ | 0.73 | 2.41 | 616.53 | 50.10 |
| ✓ | × | × | | $\langle 1, 3, 5 \rangle$ | **0.66** | **2.30** | **295.02** | 53.65 |
| ✓ | × | × | Group Size | $\langle 1, 1, 1, 1 \rangle$ | 0.66 | 2.30 | 295.02 | 53.65 |
| ✓ | × | × | | $\langle 1, 2, 2, 4 \rangle$ | 0.75 | 2.33 | 291.18 | 53.95 |
| ✓ | × | × | | $\langle 2, 4, 4, 8 \rangle$ | 0.89 | 2.44 | 284.83 | 54.40 |
| ✓ | × | × | | $\langle 4, 8, 8, 16 \rangle$ | 1.18 | 2.66 | 282.13 | 55.84 |
| ✓ | × | × | | $\langle 8, 16, 16, 32 \rangle$ | 1.75 | 3.11 | 279.48 | 55.14 |
| ✓ | × | × | | $\langle 16, 32, 64, 128 \rangle$ | 4.76 | 4.18 | **290.72** | 56.20 |
| ✓ | ✓ | × | Attention Depth | $\langle 2, 2, 2, 2 \rangle$ | 2.37 | 3.07 | 137.87 | 59.56 |
| ✓ | ✓ | × | | $\langle 1, 1, 1, 1 \rangle$ | 1.88 | 2.90 | 185.08 | 61.03 |
| ✓ | ✓ | × | Expansion Ratio | $\langle 4, 4, 4, 4 \rangle$ | 1.88 | 2.90 | 331.56 | 61.03 |
| ✓ | ✓ | × | | $\langle 3, 3, 2, 2 \rangle$ | 1.61 | 2.84 | 336.94 | **61.43** |
| ✓ | ✓ | ✓ | Teacher | + Texture Teacher | 1.61 | 2.84 | 336.94 | 59.64 |
| ✓ | ✓ | ✓ | Up | Unify Upsampling | 1.66 | 1.79 | 390.91 | 59.71 |
| ✓ | ✓ | ✓ | Gram | + Gram Supervision | 1.66 | **1.79** | 390.91 | **62.51** |

Table 2: Module ablation experiments on AutoPET-II. "Conv.": convolution encoder; "Trans.": transformer encoder; "SDKT.": spatially decoupled knowledge transfer. The best performance is in **red** and the second is in **blue**. Final setting is highlighted in green.

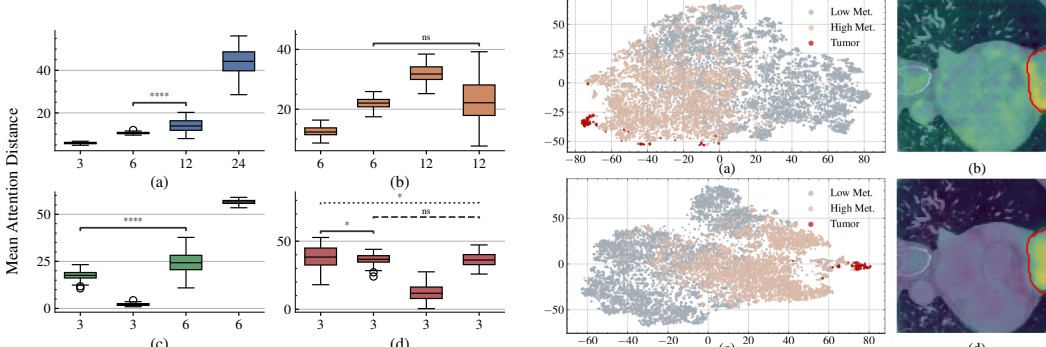

Figure 5: Average attention distribution of PWA on AutoPET-II. (a)-(d) show the results for PWA across 4 stage. Y-axis: average attention distance; X-axis: big window size. Wilcoxon rank-sum test: ns ($0.05 < p \le 1$), * ($0.01 < p \le 0.05$), **** ($p \le 0.0001$).

Figure 6: Visualization of model decoding and t-SNE plot. (a)-(b) show results without attention; (c)-(d) with attention. In (a) and (c), "Low Met." and "High Met." represent low/high-metabolism PET regions. In (b) and (d), CT background with red tumor outline.

### 3.4 MODULE ABLATION

We evaluate the performance of three model designs on AutoPET-II: JLC, PWA, and SDKT (Table 2). When using JLC alone, Params and Dice have the lowest performance. Although the framework is the simplest, the FLOPs/throughput is suboptimal due to the use of transposed convolution for upsampling. After adding the attention mechanism, the accuracy increased by 5.59%, but the throughput decreased by 233.6 Patches/s. After changing the upsampling strategy, FLOPs are significantly reduced from 2.84 G to 1.79 G, and the GPU throughput is increased from 336.94 to 599.06 Patches/s. The last three rows in the table show that it is not enough to just optimize the encoder's detail representation after adding the texture teacher. Only through the SDKT strategy based on Gram matrices can the representation ability be improved. For more specific reasons and analysis of hyper-parameter selection, please see Appendix L.

| Methods | MParams ↓ | Dice ↑ |
|---|---|---|
| FC. | 4.78 +3.58 | 79.97 +0.63 |
| FC. w. Pn. | 1.27 +0.07 | 62.00 -17.34 |
| JLC | 1.20 | 79.34 |

Table 3: Domain generalization capability comparison of the JLC and $\ell_2$ pruning methods (Filters'Importance, 2016; Fang et al., 2023) (BraTS2021 → BraTS2016 TCIA). "FC." represents full convolution, and "Pn." represents pruning operation.

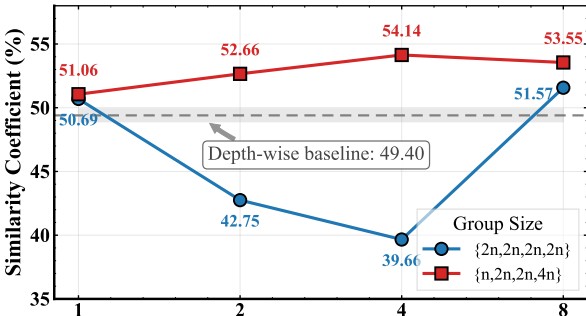

Figure 7: Dice performance comparison between different group size configurations.

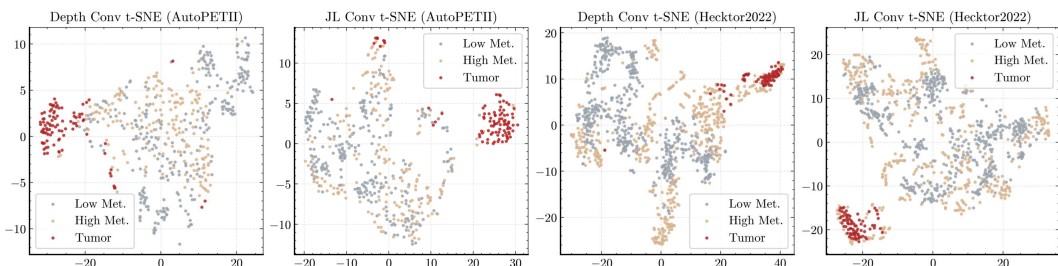

Figure 8: t-SNE plots of depth-wise convolution and JC convolution. "Low Met." and "High Met." represent low/high metabolism PET regions, respectively.

**PWA Effect Evaluation.** To verify the robustness of PWA, we conduct three experiments:

*Reduce computational redundancy through multi-scale windows.* VeloxSeg utilizes PWA to parallelize the computation of multi-scale relationships and reduce redundancy. Experimental details of Figure 5 can be found in Appendix M. The inter-group differences in PWA are significant and positively correlated with the window size, indicating that redundant information is reduced and long-distance modeling is efficient. Regarding the fourth attention stage, its design is more similar to multi-head attention, retaining some redundancy.

*Changes in features after adding PWA.* We visualize the model's decoding and its t-SNE projection, as shown in Figure 6. The results indicate that PWA helps distinguish tumor regions from high-metabolic regions while producing a more compact feature distribution.

*Effectiveness in handling heterogeneous modalities.* We test various modal input combinations, whose details could be found in Appendix N. Notably, introducing modal interaction into PWA improves the Dice score by 5.75%, significantly enhancing performance robustness without significantly increasing computational costs.

**JLC Effect Evaluation.** To verify the robustness of JLC, we conduct four experiments:

*Comparison of segmentation performance between the JL setting and the standard setting with similar parameter sizes*, as shown in Figure 7. We use a uniform kernel size of 3 to ensure a general setup. The JL-guided configuration, $\{n, 2n, 2n, 4n\}$, consistently surpasses the larger setup, $\{2n, 2n, 2n, 2n\}$, in all cases. The best performance is a 54.14% Dice score, achieved when $n = 4$. This suggests the JL-guided group size arrangement enables more robust feature extraction in a lightweight model.

*Comparison of external test performance between the JL setting and the pruned setting with similar parameter sizes*, as shown in Table 3. It verifies the generalization advantage of our lightweight convolution over the pruning method. The model with $\ell_2$ pruning (Filters'Importance, 2016) on full convolution performs significantly worse than JLC on BraTS2015 TCIA cases, even after a cycle of training, pruning, and retraining.

| Strategy | Dice ↑ | HD95 ↓ |
|---|---|---|
| − | 59.71 | 291.81 |
| +$\ell_1$ | 1.67 -58.04 | 626.79 |
| +Affinity | 41.44 -18.27 | 354.01 |
| +Shared ROI | 57.15 -2.56 | 397.43 |
| +SDKT | 62.51 +2.80 | 241.08 |

Table 4: Comparison of knowledge transfer paths constructed with different losses: Sang et al. (2021) use $\ell_1$ loss, Wang et al. (2021) use Affinity loss, Qiu et al. (2023) use agent loss in shared ROI, and we use SDKT.

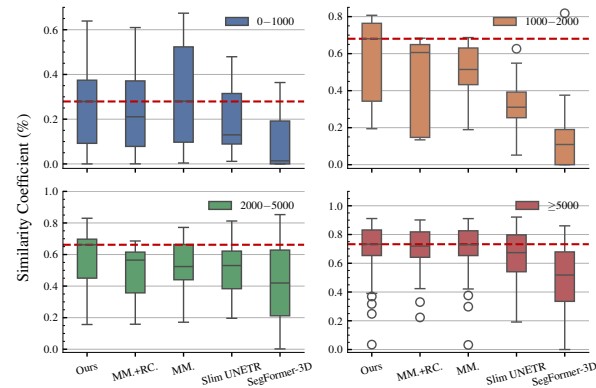

Figure 9: Box plots of Dices at different lesion volumes. "MM.": PWA+JLC multimodal backbone; "MM.+RC.": backbone with reconstruction teacher; "Ours": backbone with the teacher and SDKT.

*Testing the segmentation performance of JLC on two other datasets*, as detailed in the Appendix O. We test the segmentation performance of pure convolution networks. The convolution with JL-guided group sizes consistently outperforms the depth-wise convolution, achieving performance gains of 6.25% on Hecktor2022 and 1.16% on BraTS2021, with only a marginal increase of 0.091 million parameters. Notably, on the Hecktor2022 dataset, the JLC even surpasses the segmentation performance of the full convolution while using 0.63 million fewer parameters.

*Comparison of the t-SNE projection visualizations of JLC and depth-wise convolution* is shown in Figure 8. We test the depth-wise convolution and JLC in Figure 7 and Appendix O, providing direct visual evidence that depth-wise convolution disrupts the geometric adjacency between tokens.

**Gram-Based Transfer Effect Evaluation.** Our method is the only one to demonstrate positive knowledge transfer, as shown in Table 4. This is due to our method's avoidance of irrelevant features from the texture teachers, leading to better convergence. To further evaluate the effectiveness of SDKT, we analyze Dice across varying lesion volumes, as shown in Figure 9. VeloxSeg outperforms other lightweight models in segmentation mask fineness across all lesion volumes. Notably, for lesion volumes between 1000 and 5000, "MM.", "MM.+RC.", and "Ours" show a significant upward trend. We attribute this to the increased influence of complex textures in tumor segmentation at these sizes. The above experiments show that there is potential for optimization in small lesion segmentation. The loss weight hyperparameter experiment can be found in Appendix L.

## 4 CONCLUSION

In this paper, we propose VeloxSeg, a lightweight, theory-based framework that systematically alleviates the "*efficiency / robustness conflict*" in 3D medical image segmentation. By extending the Johnson-Lindenstrauss lemma to the convolution setting, we derive a theoretical lower bound on the group size of convolution per stage, ensuring spatial adjacency and enabling robust detail extraction. Our paired window attention mechanism, by ensembling a tumor localization team composed of attention at different scales, has near-linear complexity and more powerful modeling capabilities. Furthermore, the multimodal interaction of PWA significantly enhances model representation. Furthermore, our Spatially Decoupled Knowledge Transfer strategy establishes a positive knowledge transfer path between the self-supervised texture teacher and the segmentation network, enabling detailed representations that surpass baseline models without increasing inference overhead. Comprehensive evaluation on four diverse clinical datasets demonstrates that VeloxSeg achieves strong robustness with minimal computational cost, requiring only a single CPU core.

## 5 ACKNOWLEDGMENTS

This work was supported by the National Natural Science Foundation of China under Grant 624B2137.

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

APPENDIX

**Table of content:**

## A    USE OF LARGE LANGUAGE MODELS (LLMS)

To enhance the quality and readability of this manuscript, we use Large Language Models (LLMs) for assistance with the following tasks:

1. **Table Formatting**: Improving the presentation of tables, including adjustments to spacing, typography, and alignment to conform to publication standards.
2. **Proofreading**: Identifying and correcting grammatical errors, such as improper tense and word usage.
3. **Language Refinement**: Refining phrasing and sentence structure to improve clarity, conciseness, and overall flow.

## B    NECESSITY OF MULTI-SCALE ATTENTION

CT scans the human body using X-rays and reconstructs a two-dimensional image from one-dimensional projection data. These two-dimensional images are then stacked into a continuous three-dimensional image. CT imaging is characterized by high resolution, low tumor specificity, and rich structural information (Kramer et al., 1989).

PET generally refers to 18F-FDG PET. Radiologists use the short-lived radionuclide 18F to label glucose. After injecting this labeled glucose into the body, they observe the accumulation of glucose, which indirectly reflects the metabolic activity of human tissues. Because tumors require large amounts of glucose to support their growth and proliferation, tumor areas often appear bright in PET images (Mu et al., 2020; Kapoor et al., 2004; Bussink et al., 2011).

Magnetic resonance imaging (MRI) provides rich complementary information for analyzing brain tumors and is routinely used in clinical practice. Specifically, for gliomas, commonly used MRI sequences include T1-weighted (T1), contrast-enhanced T1-weighted (T1Gd), T2-weighted (T2), and T2 fluid-attenuated inversion recovery (T2-FLAIR) images; each sequence plays a different

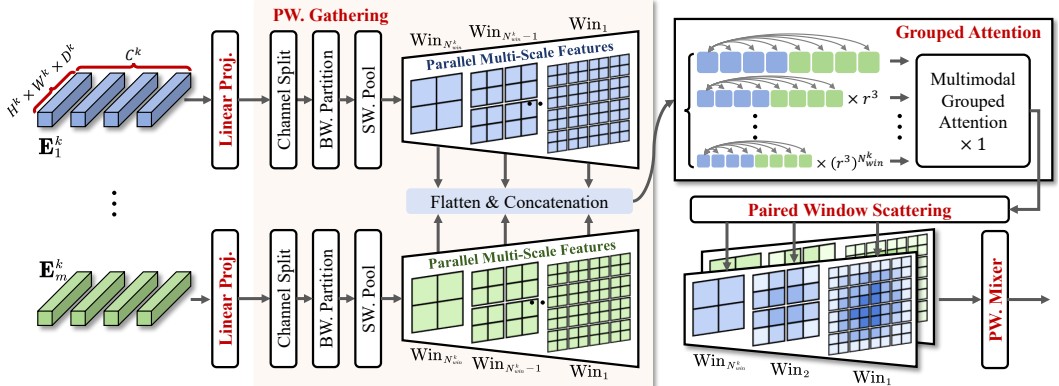

Figure 10: Detailed architecture of Paired Window Attention (PWA). This figure focuses on visually showing the feature flows of PWA.

role in distinguishing between the tumor, peritumoral edema, and the tumor core. For meningiomas, these sequences exhibit distinct characteristic features on T1Gd and contrast-enhanced T2-FLAIR (FLAIR-C) MRI images (Menze et al., 2014b; Manara et al., 2011; Kaittanis et al., 2014; Bakas et al., 2017).

This indicates that different medical modalities exhibit significant differences in their regions of interest. In tumor imaging, PET imaging, characterized by high metabolic sensitivity and low resolution, excels at localizing tumors at large scales, but its low resolution prevents clear delineation of tumor morphology. While CT imaging is less sensitive for tumors, it excels at clearly delineating tumor tissue contours at small scales. Furthermore, the four contrast types in MRI contribute differently to the identification of targets at three different scales: tumor, peritumoral edema, and tumor core. Therefore, multi-scale modality interaction is crucial in multimodal medical tasks.

## C    DETAILS OF PWA

### C.1    PYTORCH CODE

We've organized the PyTorch code and feature shape changes of PWA to help readers understand its key operations. As shown in the Algorithm 1, $N_{win} = \log(H/h_b)/\log(r) + 1$, which means that we expand the large window $(h_b, w_b, d_b)$ by $N_{win} - 1$ to obtain full-image-sized features. In the AutoPET-II dataset, we set the minimum large window size of each stage to $\langle 3, 3, 3 \rangle, \langle 6, 6, 6 \rangle, \langle 3, 3, 3 \rangle, \langle 3, 3, 3 \rangle$, which means that after the synchronous expansion of the paired windows, the large window sizes of each stage are:

- First Stage: $\langle 3, 3, 3 \rangle, \langle 6, 6, 6 \rangle, \langle 12, 12, 12 \rangle, \langle 24, 24, 24 \rangle$;
- Second Stage: $\langle 6, 6, 6 \rangle, \langle 12, 12, 12 \rangle$;
- Third Stage: $\langle 3, 3, 3 \rangle, \langle 6, 6, 6 \rangle$;
- Forth Stage: $\langle 3, 3, 3 \rangle$.

The settings of BraTS2021 are the same. In the Hecktor2022 dataset, the minimum maximum window size at each stage is $\langle 4, 4, 2 \rangle, \langle 8, 8, 4 \rangle, \langle 4, 4, 2 \rangle, \langle 4, 4, 2 \rangle$. The number of windows must be divisible by the number of channels of the feature map at the current stage to avoid extensive output channels during linear mapping. Therefore, the minimum maximum window size in the second stage is doubled.

### C.2    FEATURE FLOW

As shown in Figure 10, given the $m$-th modal feature of the $k$-th encoder stage, $\mathbf{E}_m^k \in \mathbb{R}^{C^k \times H^k \times W^k \times D^k}$, we need to first compute a set of ordered paired window sizes $\{Win_i^k\}_{i=1}^{N_{win}^k}$,

---

**Algorithm 1:** Pytorch Code of Paired Window Attention (PWA)

---

**Input:**
- **E**: Input tensor of shape $[M, C, H, W, D]$.
- $B$: Min big window size, $[h_b, w_b, d_b]$.
- $S$: Min small window size, $[h_s, w_s, d_s]$.
- $r$: Expansion ratio for the paired windows.
- $\hat{C}$: Number of channels per window after linear projection.

**Output:** Attentions of all modalities **A**

**def** PWA(*E, B, S, r*)**:**

$\quad N_{win} \leftarrow \lfloor \log(H/h_b)/\log(r) \rfloor + 1$ `// the number of paired windows`

```
/* 1) Linear Projection                                          */
```
$\quad \mathbf{Q}, \mathbf{K}, \mathbf{V} \leftarrow [\text{PWC}(\text{LN}(\mathbf{E})) \text{ for } \_ \text{ in range}(3)]$

```
/* 2) Paired Window Gathering                                    */
```
$\quad$ **for** $\mathbf{X}$ *in* $[\mathbf{Q}, \mathbf{K}, \mathbf{V}]$ **do**

$\quad\quad \mathbf{X}s \leftarrow []$ `// Initialize list for window features`

$\quad\quad$ **for** $i \leftarrow 1$ **to** $N_{win}$ **do**

$\quad\quad\quad$ `// Split channels for each window feature`

$\quad\quad\quad \mathbf{X}_i \leftarrow \mathbf{X}[:, (i-1) \cdot \hat{C} : i \cdot \hat{C}, \dots]$

$\quad\quad\quad$ `// Big and small window sizes expand synchronously`

$\quad\quad\quad S_i, B_i \leftarrow r^{i-1} \cdot S, r^{i-1} \cdot B$

$\quad\quad\quad$ `// Partition with big window`

$\quad\quad\quad \mathbf{X}_i \leftarrow \text{rearrange}\left(\mathbf{X}_i, \texttt{"M Chat (Nh hb) (Nw wb) (Nd db)}\right.$
$\quad\quad\quad\quad\quad\quad\quad\quad\quad\quad \left.\texttt{-> M (Nh Nw Nd Chat) hb wb db"}\right)$

$\quad\quad\quad$ `// Collecting a salient token for each small window`

$\quad\quad\quad \mathbf{X}_i \leftarrow \text{F.max\_pool3d}(\mathbf{X}_i, S_i, S_i)$

$\quad\quad\quad$ `// Flatten spatial dims and concatenate multimodal sequences`

$\quad\quad\quad \mathbf{X}_i \leftarrow \text{rearrange}\left(\mathbf{X}_i, \texttt{"M (N Chat) nh nw nd ->}\right.$
$\quad\quad\quad\quad\quad\quad\quad\quad\quad\quad \left.\texttt{N Chat (M nh nw nd)"}\right)$

$\quad\quad\quad \mathbf{X}s.\text{append}(\mathbf{X}_i)$

$\quad\quad$ **end**

$\quad\quad$ `// Concatenate all window features`

$\quad\quad$ `//` $\mathbf{X}$`:` $\left[\sum_{i=1}^{N_{win}} N_i, \hat{C}, M \cdot L\right]$

$\quad\quad \mathbf{X} \leftarrow \text{torch.cat}(\mathbf{X}s, \dim = 0)$

$\quad$ **end**

```
/* 3) Multimodal Grouped Attention ×1                            */
```
$\quad \mathbf{A} \leftarrow \text{multihead\_attention}(\mathbf{Q}, \mathbf{K}, \mathbf{V})$

```
/* 4) Paired Window Scattering                                   */
// Inverse of gathering.
```
$\quad$ `//` $\mathbf{A}$`:` $\left[M, \hat{C}, H, W, D\right]$

$\quad \mathbf{A} \leftarrow \text{window\_scattering}(\mathbf{A})$

```
/* 5) Paired Window Mixer                                        */
```
$\quad$ `//` $\mathbf{A}$`:` $[M, C, H, W, D]$

$\quad \mathbf{A} \leftarrow \mathbf{E} + \text{Dropout}(\text{PWC}(\mathbf{A}))$

$\quad$ **return** $A$

---

where $N_{win}^k$ is the number of window pairs:

$$\left\{Win_i^k\right\}_{i=1}^{N_{win}^k} = \left\{B_i^k, S_i^k\right\}_{i=1}^{N_{win}^k} = \left\{ \begin{array}{c} \left(r^{i-1}h_b^k, r^{i-1}w_b^k, r^{i-1}d_b^k\right), \\ \left(r^{i-1}h_s^k, r^{i-1}w_s^k, r^{i-1}d_s^k\right) \end{array} \right\}_{i=1}^{N_{win}^k}, \tag{5}$$

where $r \in \mathbb{N}$ is the expansion rate (default $r = 2$), $B_i^k$ and $S_i^k$ represent the big window and small window, respectively. $h_b^k, w_b^k, d_b^k$ represent the height, width, and depth of the big window; $h_s^k, w_s^k, d_s^k$ correspond to the small window. Notably, $r$, $B_1^k$, and $S_1^k$ are closely related to the computational cost, and the specific settings for different datasets are given in Appendix C.3. Since $B_{N_{win}^k}^k$ is equal to $\left(H^k, W^k, D^k\right)$, there is no need to set $N_{win}^k$ in advance. Subsequently, the encoder features will undergo the following steps in order:

### C.2.1 LINEAR PROJECTION

We do not follow the habit of linear mapping: the number of output channels is equal to the number of input channels, but based on the JL-guided minimum head size $C_{min}^k$, the number of heads $N_{head}^k$, and the number of window pairs $N_{win}^k$, the number of output channels is $\hat{C}^k = \min\left\{nC_{min}^k, n \in \mathbb{N} : N_{win}^k N_{head}^k \left(nC_{min}^k\right) \geq C^k\right\}$, where $\hat{C}^k$ is the actual head size. The formula is as follows:

$$\mathbf{Q}_m^k = \mathrm{PWC}\left(\mathrm{LN}\left(\mathbf{E}_m^k\right)\right), \mathbf{K}_m^k = \mathrm{PWC}\left(\mathrm{LN}\left(\mathbf{E}_m^k\right)\right), \mathbf{V}_m^k = \mathrm{PWC}\left(\mathrm{LN}\left(\mathbf{E}_m^k\right)\right), \tag{6}$$

where $\mathbf{Q}, \mathbf{K}, \mathbf{V}$ represent query, key, and value respectively. $\mathrm{LN}(\cdot)$ represents layer normalization, and $\mathrm{PWC}(\cdot)$ represents point-wise convolution. For convenience, we use $\mathbf{X}$ to represent $\mathbf{Q}, \mathbf{K}, \mathbf{V}$ in Algorithm 1 and Figure 3.

### C.2.2 PAIRED WINDOW GATHERING

Synchronously expanding paired windows ensures that the sequence lengths of $\mathbf{Q}, \mathbf{K}$ and $\mathbf{V}$ remain consistent across multiple scales, which enables parallel computation. Given $\mathbf{Q}_m^k, \mathbf{K}_m^k, \mathbf{V}_m^k \in \mathbb{R}^{C^k \times H^k \times W^k \times D^k}$, where $i$ denotes the $i$-th paired window, the processing pipeline is as follows:

- **Channel Split**: This operation assigns features to their corresponding windows along the channel dimension. The feature shape becomes $\left(N_{head}^k \times \hat{C}^k \times H^k \times W^k \times D^k\right)$.

- **Big Window Partition**: This operation partitions the features into non-overlapping blocks based on the big window size $B_i^k$. The feature shape becomes $\left(n_i^k, N_{head}^k, \hat{C}^k, r^{i-1}h_b, r^{i-1}w_b, r^{i-1}d_b\right)$, where $n_i^k$ is the total number of large windows.

- **Small Window Pooling**: This operation gathers salient tokens from each small window $S_i^k$. The feature shape becomes $\left(n_i^k, N_{head}^k, \hat{C}^k, h_b^k/h_s^k, w_b^k/w_s^k, d_b^k/d_s^k\right)$.

- **Flatten and Concatenation**: These are feature reshaping operations. The new feature size is $\left(n_i^k, N_{head}^k, \hat{C}^k, ML\right)$, where $L = (h_b^k/h_s^k) \times (w_b^k/w_s^k) \times (d_b^k/d_s^k)$ is the sequence length and $M$ is the number of modalities.

The above operations are repeated for each paired window. Thanks to the synchronous expansion, the sequence length $L$ is guaranteed to be equal across different scales. Finally, we can summarize this process as follows:

$$\tilde{\mathbf{Q}}^k = \mathrm{Gather}\left(\mathbf{Q}_1^k, \cdots, \mathbf{Q}_M^k\right), \tilde{\mathbf{K}}^k = \mathrm{Gather}\left(\mathbf{K}_1^k, \cdots, \mathbf{K}_M^k\right), \tilde{\mathbf{V}}^k = \mathrm{Gather}\left(\mathbf{V}_1^k, \cdots, \mathbf{V}_M^k\right), \tag{7}$$

### C.2.3 MULTIMODAL GROUPED ATTENTION

The similarity matrix calculation formula is as follows:

$$\mathbf{S}^k = \frac{1}{\sqrt{\hat{C}^k}}\left(\tilde{\mathbf{Q}}^k\right)^T \otimes \tilde{\mathbf{K}}^k, \tag{8}$$

where $\otimes$ represents matrix multiplication. In addition, each $L \times L$ block in the similarity matrix is assigned a relative position code $\mathbf{E}_{pos}^k$ to strengthen the position relationship between voxels in the window and cross-modal voxels. The remaining attention is calculated as follows:

$$\mathbf{W}^k = \text{softmax}\left(\mathbf{S}^k + \mathbf{E}_{pos}^k\right), \tag{9}$$

$$\mathbf{A}^k = \mathbf{W}^k \otimes \hat{\mathbf{V}}^k, \tag{10}$$

$\mathbf{W}^k$ is the attention weight matrix, and $\mathbf{A}^k$ is the attention obtained for each window.

### C.2.4 PAIRED WINDOW SCATTERING

After computing the attention mechanism in parallel, we perform the inverse operation of Paired Window Gathering to map the multi-scale attention mechanism to the original feature space, obtaining the window attention $\mathbf{A}_m^k$ for each modality, which has the same size as $\mathbf{Q}_m^k, \mathbf{K}_m^k, \mathbf{V}_m^k$.

$$\mathbf{A}_1^k, \cdots, \mathbf{A}_M^k = \text{Scatter}(\mathbf{A}^k). \tag{11}$$

### C.2.5 PAIRED WINDOW MIXER

The above operations obtain window attention of different scales. We will use $1 \times 1 \times 1$ convolution to mix them to get the final feature $\tilde{\mathbf{E}}_m^k$. The formula is as follows:

$$\tilde{\mathbf{E}}_m^k = \mathbf{E}_m^k + \text{PWC}\left(\mathbf{A}_m^k\right), m = 1, \cdots, M. \tag{12}$$

### C.3 COMPUTATIONAL COMPLEXITY

Let $N = H \cdot W \cdot D$, $B = h_b \cdot w_b \cdot d_b$, $S = h_s \cdot w_s \cdot d_s$, and $\kappa = 1 + \frac{1}{r^2} + \cdots + \frac{1}{r^{2N_{win}}} = \frac{1 - r^{-3N_{win}}}{1 - r^{-3}}$, the computational complexity of PWA is calculated as follows:

$$\underbrace{\left(\frac{N}{B}\right)\left(1 + \frac{1}{r^2} + \cdots + \frac{1}{r^{2N_{win}}}\right)}_{\text{number of big windows}} \quad \underbrace{\left(4\frac{B}{S}C^2 + 2\frac{B^2}{S^2}C\right)}_{\text{multiplication operations per big window}}$$

$$= N\kappa\left(4\frac{1}{S}C^2 + 2\frac{B}{S^2}C\right)$$

$$= \left(\frac{N\kappa}{S}\right)\left(4C^2 + 2\frac{B}{S}C\right),$$

## D DETAILS OF JL-GUIDED GROUP SIZE

### D.1 EMPIRICAL PARAMETRIZATION OF COVERING NUMBERS

Motivated by the classical covering-number results in Heinonen (2001, Definition 10.15 and Exercise 10.17), we consider a hypothesis class whose covering number satisfies

$$N(\epsilon) \leq C\left(\frac{1}{\epsilon}\right)^\beta, \qquad C \geq 1, \ \epsilon \in (0, 1), \tag{13}$$

where $C \geq 1$ is a constant independent of the ball, $\epsilon \in (0, 1)$. To ensure generality, we do not make further assumptions about the data to obtain specific parameters, but instead use empirical functions for approximation: $\hat{N} = (M \cdot v)^\alpha$.

- $M \cdot v$ replaces $1/\epsilon$, representing the coverage density required per dimension. Given the constraints of JL's logarithmic scaling and the requirement that the group size divides the input channel, we omit the possible constant term.
- $\alpha$ serves as a difficulty coefficient reflecting the dataset's intrinsic complexity. We calibrate $\alpha$ based on the most challenging dataset to ensure robust generalization across different tasks.

## D.2 WHY WE AVOID MORE COMPLEX POLYNOMIAL APPROXIMATIONS.

The covering-number estimate is only an intermediate step; JL lemma then applies a logarithm. Consequently, low-degree and constant terms in a polynomial approximation have a minimal effect after taking the log. For example, consider $v = 43, M = 2, \alpha = 1$. If we add a constant term $\beta$, $\log(Mv + \beta) - \log(Mv) = \log(1 + \beta/128) \approx 0$. Besides, because the group_size must divide input_channel, such a small interrupt rarely changes the final group size.

## D.3 ANALYSIS OF DIFFERENT NUMBER OF MODALITIES AND THE GROUP SIZE OF 2D IMAGES

For a typical lightweight 3D medical segmentation method, its network contains $M \in \{1, 2, 4\}$ modalities with volume ratios $\{v^k\}_{k=1}^4 = \{4^3, 8^3, 16^3, 32^3\}$ of each stage. Its complexity increases with depth, which means that the group size is:

$$C_{\text{group}} \approx \begin{cases} \{4.2\alpha, 6.2\alpha, 8.3\alpha, 10.4\alpha\}, & M = 1 \\ \{4.9\alpha, 6.9\alpha, 9.0\alpha, 11.1\alpha\}, & M = 2 \\ \{5.5\alpha, 7.6\alpha, 9.7\alpha, 11.8\alpha\}, & M = 4 \end{cases} \tag{14}$$

Considering that the group size needs to be divisible by the total number of channels and that nonlinear networks have stronger compression capabilities than linear networks, we use $\{C^k_{\text{group}}\}_{k=1}^4 = \{4\alpha, 8\alpha, 8\alpha, 16\alpha\}$ for each stage, where $\alpha$ is determined by the most difficult AutoPET-II dataset to ensure universality. For convenience, we replace $\alpha$ with $n = \lceil \alpha/4 \rceil \in \mathbb{N}$, and the final convolution group size of each stage of the network is set to $\{C^k_{\text{group}}\}_{k=1}^4 = \{n, 2n, 2n, 4n\}$.

For lightweight convolution settings in the natural image domain, the input image typically has $M = 3$ channels, and the volume ratio of each stage of the network is $\{v^k\}_{k=1}^4 = \{1^2, 2^2, 4^2, 8^2\}$, which means the group size is: $\{C^k_{\text{group}}\}_{k=1}^4 = \{\alpha \log 3, \alpha \log 12, \alpha \log 48, \alpha \log 192\} \approx \{1.1\alpha, 2.5\alpha, 3.9\alpha, 5.3\alpha\}$. Considering the integer divisibility of the channels, it is recommended to use a group size of $\{\alpha, 2\alpha, 4\alpha, 4\alpha\}$. This setting is similar to the depth-wise convolution setting, further demonstrating the effectiveness of depth-wise convolution in the natural image domain.

# E DETAILS OF LOSS FUNCTION

## E.1 SEGMENTATION LOSS

For segmentation, we use a combination of the cross entropy loss $\mathcal{L}_{ce}$ and the foreground dice loss $\mathcal{L}_{dice}$, which can optimize the detail and global segmentation effects. Deep supervision is performed on the segmentation decoder. The formula is as follows:

$$\mathcal{L}_{ce}(\mathbf{P}, \mathbf{Y}) = -\frac{1}{HWD} \sum_{i=1}^{HWD} \mathbf{P}_i \log(\mathbf{Y}_i), \tag{15}$$

$$\mathcal{L}_{dice}(\mathbf{P}, \mathbf{Y}) = 1 - \frac{2 \sum_{i=1}^{HWD} \mathbf{P}_i \mathbf{Y}_i}{\sum_{i=1}^{HWD} \mathbf{P}_i + \sum_{i=1}^{HWD} \mathbf{Y}_i}, \tag{16}$$

where $\mathbf{P}$ is prediction map and $\mathbf{Y}$ is segmentation ground truth, subscript $i$ represents the $i$-th voxel.

## E.2 RECONSTRUCTION LOSS

The texture teacher learns without data annotation, reconstructing the original input image based on model features. The loss function is a simple mean squared error, as shown in the following formula:

$$\mathcal{L}_{rc} = \frac{1}{M} \sum_{m=1}^{M} \|\mathbf{R}_m - \mathbf{I}_m\|^2, \tag{17}$$

where $M$ is the number of input modalities, $\mathbf{R}_m$ and $\mathbf{I}_m$ represent the reconstructed and original images of the $m$-th modality, respectively.

| Dataset | Modalities | Region | Label Type | Image Size | Crop Size | Voxel Spacing |
|---------|-----------|--------|-----------|-----------|-----------|---------------|
| AutoPET-II | PET, CT | Whole Body | Malignant melanoma, lymphoma, or lung cancer lesions | Min $(400, 400, 200)$ | $(96, 96, 96)$ | Fixed $(2.036, 2.036, 3)$ |
| Hecktor2022 | PET, CT | Head & Neck | Primary gross tumor volume (GTVp), or lymph node gross tumor volume (GTVn) | Min $(128, 128, 67)$ | $(128, 128, 64)$ | Median $(0.98, 0.98, 3.3)$ |
| BraTS2021 | MRI | Brain | Brain tumors: whole tumor (WT), tumor core (TC), enhancing tumor (ET) subregions. | Fixed $(240, 240, 155)$ | $(96, 96, 96)$ | Fixed $(1.0, 1.0, 1.0)$ |
| BraTS 2016 TCIA | MRI | Brain | Glioma segmentation (multi-class): necrosis/active tumor and edema. | Fixed $(240, 240, 155)$ | $(96, 96, 96)$ | Fixed $(1.0, 1.0, 1.0)$ |

Table 5: Details of AutoPET-II, Hecktor2022, BraTS2021, and BraTS 2016 TCIA datasets. If image size is a variable, the minimum value is reported. If voxel spacing is a variable, the median value is reported.

## F   DETAILS OF DATASET

We evaluate our proposed VeloxSeg on four public medical image datasets: AutoPET-II (Gatidis S, 2022), Hecktor2022 (Oreiller et al., 2022), BraTS2021 (Baid et al., 2021), and BraTS 2016 TCIA (Menze et al., 2014a). The AutoPET-II and Hecktor2022 datasets are multimodal PET/CT datasets for tumor segmentation. AutoPET-II contains 1,014 whole-body PET/CT scans with variable image dimensions and is cropped to $96 \times 96 \times 96$ patches. Hecktor2022 comprises 524 head and neck PET/CT scans, cropped to $128 \times 128 \times 64$ patches. The BraTS2021 and BraTS 2016 TCIA datasets are multimodal (T1, T1ce, T2, FLAIR) MRI datasets for brain tumor segmentation. Each patient's data is registered to a common spatial resolution of $240 \times 240 \times 155$ and undergoes skull stripping. The brain tumor region is segmented into three primary sub-regions: the enhancing tumor (ET), the tumor core (TC), and the whole tumor (WT). For training efficiency, volumes are cropped to $96 \times 96 \times 96$ patches. BraTS2021 contains 1,251 cases for training and validation, while BraTS 2016 TCIA (244 cases) serves as an external test set to evaluate domain generalization capability across different data distributions.

We use four public medical image datasets to verify the effectiveness of VeloxSeg, including AutoPET-II, Hecktor2022, BraTS2021, and BraTS 2016 TCIA, which is used as an external test set to compare the generalization ability of the model. The first two datasets contain CT and PET images, and the latter two datasets contain 3D MRI images with four modalities. The details of the datasets are described in Table 5.

## G   DETAILS OF COMPUTATIONAL PERFORMANCE

We evaluate the computational performance of VeloxSeg against other leading models on the AutoPET-II, Hecktor2022, and BraTS2021 datasets. Our analysis focus on four key metrics: the number of model parameters in millions, GFLOPs, GPU throughput, and CPU throughput. On the AutoPET-II and Hecktor2022 datasets, VeloxSeg established a new standard for efficiency. As detailed in Table 6, our model operates with only 1.66 million parameters and the lowest GFLOPs among all competitors, requiring just 1.79 on AutoPET-II and 2.13 on Hecktor2022. This lean profile translates to exceptional speed, where VeloxSeg recorded the highest GPU throughput and second-highest CPU throughput on both datasets. In the lightweight category, while Slim UNETR is marginally smaller, VeloxSeg surpasses it in computational cost and processing speed.

| Methods | Type | AutoPET-II | | | | Hecktor2022 | | | |
|---|---|---|---|---|---|---|---|---|---|
| | | MP. | GF. | ThrG. | ThrC. | MP. | GF. | ThrG. | ThrC. |
| **Basic Models** | | | | | | | | | |
| UNet | CNN | 5.75 | 136.56 | 101.04 | 0.23 | 5.75 | 161.84 | 85.60 | 0.19 |
| VNet | CNN | 45.60 | 322.22 | 58.99 | 0.14 | 45.60 | 381.89 | 49.82 | 0.11 |
| MedNeXt-S | CNN | 5.54 | 57.93 | 27.95 | 0.06 | 5.54 | 68.54 | 23.20 | 0.05 |
| UNETR | CNN-Transformer | 95.76 | 83.61 | 131.96 | 0.40 | 95.76 | 99.09 | 105.78 | 0.35 |
| Swin UNETR | CNN-Transformer | 15.51 | 84.26 | 38.37 | 0.14 | 15.51 | 100.66 | 28.58 | 0.10 |
| VSmTrans | CNN-Transformer | 12.48 | 91.44 | 36.56 | 0.14 | 3.12 | 28.79 | 34.13 | 0.16 |
| UNETR++ | CNN-Transformer | 19.97 | 57.93 | 161.15 | 0.67 | 19.97 | 68.66 | 138.39 | 0.56 |
| U-KAN | CNN-KAN | 7.06 | 22.90 | 187.06 | 0.82 | 7.06 | 27.13 | 159.92 | 0.75 |
| **Multimodal Models** | | | | | | | | | |
| Nestedformer | CNN-Transformer | 4.71 | 58.62 | 95.63 | 0.41 | 4.71 | 69.48 | 79.05 | 0.35 |
| A2FSeg | CNN | 41.32 | 207.97 | 52.02 | 0.17 | 41.32 | 246.48 | 40.60 | 0.13 |
| H-DenseFormer | CNN-Transformer | 3.64 | 71.91 | 123.35 | 0.44 | 3.64 | 85.23 | 102.80 | 0.37 |
| **Lightweight Models** | | | | | | | | | |
| SegFormer-3D | CNN-Transformer | 4.50 | 5.11 | **364.24** | 3.31 | 4.50 | 6.06 | **305.01** | 2.67 |
| Slim UNETR | CNN-Transformer | 1.77 | **3.83** | 178.33 | **11.40** | 1.77 | **4.53** | 151.85 | **8.78** |
| SuperLightNet | CNN-Transformer | 2.75 | 19.42 | 55.48 | 0.27 | 2.75 | 23.01 | 47.36 | 0.23 |
| HCMA-UNet | CNN-Mamba | 2.81 | 26.15 | 54.51 | – | 2.81 | 31.00 | 46.48 | – |
| U-RWKV | CNN-RWKV | **1.44** | 20.68 | 82.09 | – | **1.44** | 24.51 | 73.98 | – |
| VeloxSeg | CNN-Transformer | **1.66** | **1.79** | **390.91** | **6.67** | **1.66** | **2.13** | **319.80** | **5.47** |

Table 6: Computational performance comparison of all models on AutoPET-II and Hecktor2022 datasets. "MP.": Million Parameters; "GF.": GFLOPs; "ThrG.": Throughput on GPU; "ThrC.": Throughput on CPU.

On the BraTS2021 dataset, we test early-fusion VeloxSeg due to its concentrated target distribution, absence of small lesions, and low modality heterogeneity. Table 7 shows that VeloxSeg-C is one of the smallest models with only 1.46 million parameters, yet it achieves the lowest GFLOPs at 2.64. Most notably, it delivered the highest GPU throughput of any model, processing 536.62 images/s, alongside the second-fastest CPU throughput. This positions VeloxSeg-C as a more efficient and faster alternative to other lightweight models like U-RWKV and SegFormer-3D.

Across all three benchmarks, the VeloxSeg architecture demonstrates an excellent balance between model size, computational requirements, and processing speed, making it well-suited for deployment in resource-constrained environments. Furthermore, segmentation methods based on sequence models, such as Mamba and RWKV, lack CPU support, significantly limiting their application in edge devices.

# H   DETAILS OF GPU MEMORY USAGE

Table 8 shows that VeloxSeg has the second lowest memory footprint, saving more GPU memory than all non-lightweight baseline models. As shown in Figure 11, on the AutoPET-II dataset, the base methods' GPU memory usage is 2.5 to 20 times that of VeloxSeg, with inference memory usage reaching up to 24 times higher. Compared to other lightweight models, VeloxSeg consistently has less GPU memory usage than SegFormer-3D, SuperLightNet, HCMA-UNet, and U-RWKV, which, despite claiming to be lightweight, have GPU memory usage that is 5.9 to 17 times higher than ours.

| Methods | Type | MP. ↓ | GF. ↓ | ThrG. ↑ | ThrC. ↑ |
|---|---|---|---|---|---|
| **Basic Models** | | | | | |
| UNet | CNN | 5.75 | 138.14 | 103.70 | 0.23 |
| VNet | CNN | 45.61 | 322.85 | 63.37 | 0.14 |
| MedNeXt-S | CNN | 5.54 | 57.95 | 27.75 | 0.06 |
| UNETR | CNN-Transformer | 102.06 | 85.79 | 128.16 | 0.41 |
| Swin UNETR | CNN-Transformer | 15.51 | 85.53 | 38.11 | 0.13 |
| VSmTrans | CNN-Transformer | 12.48 | 92.72 | 36.18 | 0.13 |
| UNETR++ | CNN-Transformer | 19.98 | 58.81 | 153.63 | 0.52 |
| U-KAN | CNN-KAN | 7.06 | 23.69 | 181.89 | 0.85 |
| **Multimodal Models** | | | | | |
| Nestedformer | CNN-Transformer | 7.52 | 88.43 | 56.92 | 0.23 |
| A2FSeg | CNN | 74.55 | 361.18 | 28.92 | 0.08 |
| H-DenseFormer | CNN-Transformer | 5.39 | 73.18 | 95.61 | 0.42 |
| **Lightweight Models** | | | | | |
| SegFormer-3D | CNN-Transformer | 4.53 | **5.42** | **355.73** | 2.98 |
| Slim UNETR | CNN-Transformer | 1.78 | 6.59 | 97.50 | **10.62** |
| SuperLightNet | CNN-Transformer | 2.75 | 19.54 | 55.13 | 0.28 |
| HCMA-UNet | CNN-Mamba | 2.81 | 26.69 | 53.72 | — |
| U-RWKV | CNN-RWKV | **1.43** | 21.08 | 83.15 | — |
| VeloxSeg-C | CNN-Transformer | **1.46** | **2.64** | **536.62** | **5.23** |

Table 7: Computational performance on BraTS2021 dataset with patch size $96 \times 96 \times 96$ and $4$ modalities (T1/T1ce/T2/FLAIR). "MP.": Million Parameters; "GF.": GFLOPs; "ThrG.": Throughput on GPU; "ThrC.": Throughput on CPU.

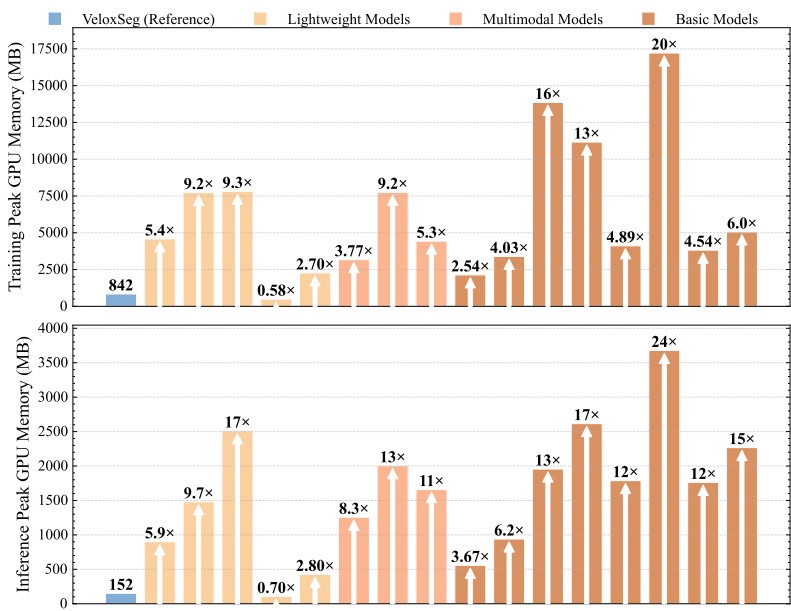

Figure 11: Visualization of memory usage for training and inference of all models. Arranged from left to right in reverse order of Table 6.

| Methods | Type | AutoPET-II | | Hecktor2022 | | BraTS2021 | |
|---------|------|:----:|:----:|:----:|:----:|:----:|:----:|
| | | TM. | IM. | TM. | IM. | TM. | IM. |
| **Basic Models** | | | | | | | |
| UNet | CNN | 5054 | 2268 | 5942 | 2698 | 5112 | 2090 |
| VNet | CNN | 3820 | 1762 | 4460 | 2072 | 3886 | 1684 |
| MedNeXt | CNN | 17216 | 3678 | 20372 | 4376 | 17258 | 3734 |
| UNETR | CNN-Transformer | 4114 | 1788 | 4626 | 1914 | 4106 | 1460 |
| Swin UNETR | CNN-Transformer | 11164 | 2616 | 13258 | 3704 | 11166 | 3466 |
| VSmTrans | CNN-Transformer | 13856 | 1956 | 13318 | 2532 | 14202 | 1972 |
| UNETR++ | CNN-Transformer | 3392 | 940 | 3906 | 1088 | 3640 | 984 |
| U-KAN | CNN-KAN | 2138 | 558 | 2360 | 640 | 2200 | 606 |
| **Multimodal Models** | | | | | | | |
| NestedFormer | CNN-Transformer | 4428 | 1658 | 5182 | 1966 | 7274 | 2200 |
| A2FSeg | CNN | 7748 | 2004 | 9102 | 2352 | 15604 | 3074 |
| H-DenseFormer | CNN-Transformer | 3172 | 1256 | 3778 | 1474 | 3712 | 1094 |
| **Lightweight Models** | | | | | | | |
| SegFormer-3D | CNN-Transformer | 2272 | 426 | 2702 | 436 | 2096 | **430** |
| Slim UNETR | CNN-Transformer | **488** | **106** | **562** | **124** | **602** | **172** |
| SuperLightNet | CNN-Transformer | 7812 | 2510 | 9192 | 2730 | 7842 | 2266 |
| HCMA-UNet | CNN-Mamba | 7730 | 1480 | 9098 | 1892 | 7938 | 2020 |
| U-RWKV | CNN-RWKV | 4582 | 900 | 5388 | 1044 | 4670 | 936 |
| VeloxSeg | CNN-Transformer | **842** | **152** | **1006** | **178** | **1392** | 1112 |

Table 8: Peak GPU memory usage for training and inference for all models across three datasets. All models were tested with a fixed batch size of 2, ensuring all other experimental conditions remained the same. "TM." represents the peak GPU memory usage during training, and "IM." represents the peak GPU memory usage during inference.

| Dataset | Model | MParams $\downarrow$ | GFLOPs $\downarrow$ | Thr.GPU $\uparrow$ | Thr.CPU $\uparrow$ | Dice $\uparrow$ | HD95 $\downarrow$ |
|---------|-------|:----:|:----:|:----:|:----:|:----:|:----:|
| AutoPET-II | nnUNet | 88.62 | 3078.83 | 81.13 | 0.127 | 55.85 | 193.54 |
| | VeloxSeg | **1.66** | **1.79** | **390.91** | **6.67** | **70.05** | **177.51** |
| Hecktor2022 | nnUNet | 88.62 | 4828.04 | 68.02 | 0.106 | 60.80 | 36.67 |
| | VeloxSeg | **1.66** | **2.13** | **319.80** | **5.47** | **62.51** | **30.22** |

Table 9: Performance comparison between nnUNet and VeloxSeg across PET/CT datasets. Both segmentation performance and computational efficiency are evaluated.

# I   RESULTS ON THE NNUNET TRAINING FRAMEWORK

To unleash more model potential, we placed the model in the nnUNet training framework and completed the training while keeping the patch size consistent with the experimental setting. The results are shown in Table 9. It can be seen that our model has achieved comprehensive transcendence. In the AutoPET-II dataset, we achieved a Dice that was 14.2% higher than the nnUNet baseline with 1.87% of the parameters and 5.81e-2% GLOPs, and the GPU throughput and CPU throughput increased by 4.8$\times$ and 52.5$\times$ respectively. Similarly, in the Hecktor2022 dataset, VeloxSeg achieved a Dice that was 1.71% higher than the nnUNet baseline with 1.87% of the parameters and 4.41e-2% GLOPs, and the GPU throughput and CPU throughput increased by 4.7$\times$ and 51.6$\times$ respectively.

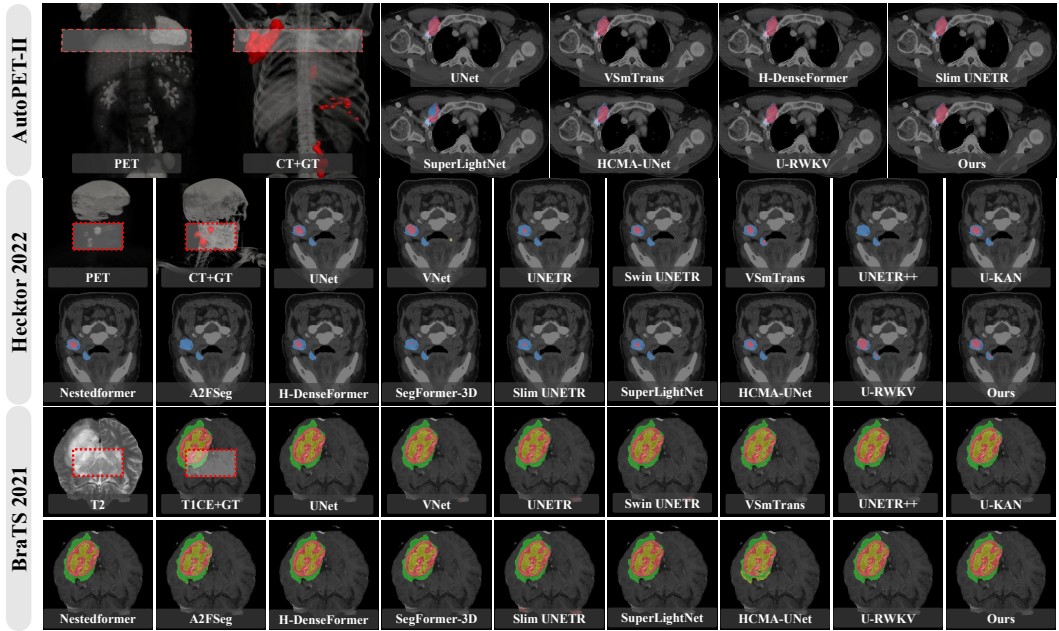

Figure 12: 3D qualitative visualization of different methods on the AutoPET-II, Hecktor2022 and BraTS2021 datasets. In PET-CT datasets, model predictions are shown on CT images (red indicates true positives, yellow indicates false positives, and blue indicates false negatives); In MRIs datasets, model predictions are shown on T1CE images (red represents ET, yellow represents TC, and green represents WT).

## J  QUALITATIVE RESULTS

Figure 12 show the qualitative results of three segmentation cases: large melanoma lesions, primary and secondary lesions of right tonsil tumors or lymphomas, and glioma lesions. It can be seen that our method can accurately locate the tumor area, exclude the wrong attention to areas such as intracranial veins, and the prediction results are consistent with the labels.

## K  MODALITY ADAPTATION EVALUATION

On BraTS2021 MRI brain tumor dataset, we use an early fusion strategy (VeloxSeg-C) that does not slow down as the number of modalities increases, as shown in Table 10. Since brain tumors are large and centralized, and the slices processed are relatively fixed, almost all models achieved good results. Our model improves the Dice by 1.72% compared to the state-of-the-art SuperLightNet.

## L  HYPERPARAMETER ANALYSIS

### L.1  MODULE HYPERPARAMETER

Model optimization, detailed in Table 2, focused on balancing segmentation performance and computational efficiency. convolution adjustments, including reducing model width $\langle 32, 61, 128, 256 \rangle$ to $\langle 16, 32, 64, 128 \rangle$, improved CPU throughput, from 10.83 to 20.23, and Dice, from 48.96% to 50.10%. Replacing large kernel convolution $\langle 7 \rangle$ with parallel small kernels $\langle 1, 3, 5 \rangle$ yielded a 3.55% Dice increase, from 50.10% to 53.65%, while simultaneously reducing MParams from 0.73 to 0.66, and GFLOPs from 2.41 to 2.30. Optimal group channel setting $\langle 4, 8, 8, 16 \rangle$ achieved a 55.14% Dice. Attention depth reduction $\langle 2, 2, 2, 2 \rangle$ to $\langle 1, 1, 1, 1 \rangle$ surprisingly enhanced Dice, from 59.56% to 61.03%. As suggested in Table 11, reducing the FFN dilation rate of Transformer/Convolution to $\langle 3, 3, 2, 2 \rangle$ can slightly improve Dice performance while reducing computational cost.

| Method | Dice Similarity Coefficient (%) | | | | Hausdorff Distance 95% (mm) | | | |
|---|---|---|---|---|---|---|---|---|
| | Avg. ↑ | ET ↑ | TC ↑ | WT ↑ | Avg. ↓ | ET ↓ | TC ↓ | WT ↓ |
| UNet | 88.18 | 89.62 | 85.65 | 89.28 | 4.93 | 5.95 | 7.52 | 3.59 |
| V-Net | 88.86 | 90.66 | 86.16 | 89.75 | 5.26 | 5.98 | 8.29 | 4.31 |
| MedNeXt-S | 90.70 | 92.64 | 88.33 | 91.12 | 4.48 | 4.72 | 7.10 | 3.33 |
| UNETR | 85.44 | 88.12 | 81.49 | 86.71 | 6.68 | 8.51 | 8.19 | 4.75 |
| Swin UNETR | 88.52 | 90.19 | 85.73 | 89.63 | 5.07 | 6.37 | 7.29 | 3.44 |
| VSmTrans | 86.62 | 91.15 | 78.01 | 90.71 | 7.00 | 6.00 | 12.07 | 3.44 |
| UNETR++ | 88.77 | 90.26 | 87.01 | 89.05 | 4.49 | 5.30 | 6.98 | 3.76 |
| U-KAN | 88.51 | 90.57 | 86.14 | 88.82 | 5.44 | 5.77 | 8.03 | 4.65 |
| Nestedformer | 88.54 | 89.60 | 86.71 | 89.30 | **4.21** | 5.44 | 7.38 | **3.13** |
| A2FSeg | 88.18 | **91.78** | 84.58 | 88.18 | 4.66 | **4.47** | 7.51 | 3.72 |
| H-DenseFormer | 89.35 | 90.80 | 86.66 | 90.59 | 5.58 | 5.85 | 8.88 | 3.97 |
| SegFormer-3D | 89.18 | 90.37 | **87.49** | 89.69 | 4.61 | 5.45 | **6.30** | 3.57 |
| Slim UNETR | 87.33 | 89.31 | 85.00 | 87.66 | 5.16 | 6.46 | 7.57 | 3.52 |
| SuperLightNet | **89.72** | 91.46 | 87.22 | 90.48 | 4.46 | 5.39 | 6.33 | **3.06** |
| HCMA-UNet | 89.53 | 91.63 | 86.24 | **90.72** | 4.79 | 5.05 | 8.12 | 4.15 |
| U-RWKV | 89.04 | 91.34 | 86.94 | 88.83 | 5.42 | 6.39 | 7.66 | 4.63 |
| VeloxSeg-C | **91.44** | **93.09** | **89.00** | **92.24** | **3.75** | **3.89** | **4.41** | 3.35 |

Table 10: Segmentation performance comparison on the BraTS2021 dataset. VeloxSeg-C's metrics are highlighted in green . The best performance is red and the second best performance is blue.

| Depth | CT | PET | Enc. | Dec. |
|---|---|---|---|---|
| 1 | 3.07 | 3.22 | 3.02 | 3.06 |
| 2 | 2.85 | 2.82 | 2.73 | 2.75 |
| 3 | 2.37 | 2.38 | 2.26 | 2.29 |
| 4 | 1.82 | 1.85 | 1.74 | − |

Table 11: Ratio of channels to input embeddings after pruning the FFN layers of PWA and JLC. Baseline Dice: 69.94%; after pruning: 68.49%.

| Ablation | $\lambda_{rc}$ | $\lambda_{sdkt}$ | Dice ↑ |
|---|---|---|---|
| $\mathcal{L}_{rc}$ | 1.5 | 1.5 | 58.03 -4.41 |
| | 1.0 | 1.5 | 61.53 -0.92 |
| | 0.5 | 1.5 | 62.44 |
| $\mathcal{L}_{style}$ | 0.5 | 2.5 | 62.23 -0.28 |
| | 0.5 | 2.0 | 62.51 |
| | 0.5 | 1.5 | 62.44 -0.07 |
| | 0.5 | 1.0 | 61.66 -0.85 |
| | 0.5 | 0.5 | 60.55 -2.06 |

Table 12: Hyperparameters experiments with loss weight on AutoPET-II.

## L.2 LOSS WEIGHT

Considering the differences in the contributions of various tasks to segmentation, we need to adjust the relative weights between different tasks to explore the optimal parameter update process. To this end, we adopt the strategy of controlling variables and adjust the loss weights of $\mathcal{L}_{rc}$ and $\mathcal{L}_{style}$ in turn. The specific results are shown in Table 12. The final parameters of each experiment are highlighted in green , and red and green are used to indicate the improvement and deterioration in the process. Finally, the optimal weight parameters are selected as $\lambda_{rc} = 0.5$, $\lambda_{sdkt} = 2.0$.

## M COMPUTATION OF MEAN ATTENTION DISTANCE

To analyze the locality of attention heads, we compute the Mean Attention Distance (MAD), a metric that measures the average physical distance between a query voxel and the key voxels it attends to, weighted by the attention scores. We extend the Mean Attention Distance metric to 3D volumes to analyze attention patterns in volumetric data, and its computation is detailed below.

| Modality | MParams ↓ | GFLOPs ↓ | Thr. (GPU) ↑ | Dice ↑ |
|---|---|---|---|---|
| CT | 1.39 -0.27 | 1.61 -0.18 | 768.48 +377.57 | 21.43 -41.01 |
| PET | 1.39 -0.27 | 1.61 -0.18 | 768.48 +377.57 | 49.28 -13.16 |
| PET + CT | 1.39 -0.27 | 1.70 -0.09 | 694.76 +303.85 | 56.69 -5.75 |
| ⟨CT, PET⟩ | 1.66 | 1.79 | 390.91 | 62.51 |

Table 13: Modality ablation experiments performed on AutoPET-II. "PET+CT" indicates an early fusion strategy, and "⟨CT, PET⟩" indicates consideration of modality interaction.

Let the input 3D volume be partitioned into a grid of $H \times W \times D$ voxels, resulting in a total of $L = H \cdot W \cdot D$ voxels. The attention weight matrix for a given head is denoted by $\mathbf{W} \in \mathbb{R}^{L \times L}$, where $\mathbf{W}_{ij}$ is the attention weight from voxel $i$ to voxel $j$. Let $\mathbf{s} \in \mathbb{R}^+$ be a scalar representing the physical edge length of a single cubic voxel.

First, we map each voxel's flattened 1D index $i$ (where $i \in \{0, \ldots, L-1\}$) to its physical 3D coordinates $(x_i, y_i, z_i)$. Assuming a standard row-major flattening order where the x-axis (width) is the fastest-changing dimension and the z-axis (depth) is the slowest:

$$\begin{cases} z_i = \lfloor i / (H \cdot W) \rfloor \\ y_i = \lfloor (i \bmod (H \cdot W)) / W \rfloor \\ x_i = i \bmod W \end{cases} . \tag{18}$$

Next, we compute the pairwise physical distance matrix $\mathbf{D} \in \mathbb{R}^{L \times L}$. Each element $\mathbf{D}_{ij}$ represents the scaled Euclidean distance between the centers of voxel $i$ and voxel $j$:

$$\mathbf{D}_{ij} = \mathbf{s} \cdot \sqrt{(x_i - x_j)^2 + (y_i - y_j)^2 + (z_i - z_j)^2} \tag{19}$$

Finally, the Mean Attention Distance (MAD) is formulated as the expectation of the physical distance over the attention distribution. As shown in Equation 20, it is computed by summing all distances weighted by their corresponding attention scores, and then averaging over all query voxels:

$$\text{MAD} = \frac{1}{L} \sum_{i=0}^{L-1} \sum_{j=0}^{L-1} \mathbf{W}_{ij} \cdot \mathbf{D}_{ij} \tag{20}$$

A smaller MAD value indicates that the attention head primarily focuses on local information, whereas a larger MAD value signifies a more global attention pattern across the volume. The window size of each stage of PWA is shown in Appendix C.

# N  PWA MULTIMODAL EVALUATION

To verify the effectiveness of PWA in heterogeneous modal modeling, we test various inputs, as shown in Table 13. Using only PET or CT reduces model size and complexity but sacrifices segmentation performance. An early fusion strategy achieves a Dice of 56.69%, outperforming the pure convolution framework's 55.84%. Crucially, introducing modal interaction in PWA improves Dice by 5.75%, significantly improving performance robustness without significantly increasing computational or time costs.

# O  JL-SETTING GENERALIZATION EVALUATION

Results demonstrate the effectiveness of JL-guided group size configurations $\langle 4, 8, 8, 16 \rangle$ on various datasets. While the smallest configuration $\langle 1, 1, 1, 1 \rangle$ achieves the lowest computational cost, reducing segmentation performance by 0.091 MParams and 0.063 to 0.075 GFLOPs, it significantly degrades segmentation performance, particularly on the Hecktor2022 dataset, which features heterogeneous modality data and cross-organ distribution of targets, where the Dice drops by 6.25%. Larger configurations $\langle 16, 32, 64, 128 \rangle$ only slightly improve the Dice (by 1.16% on BraTS2021) but significantly increase computational complexity by 0.633 MParams and 0.268 to 0.317 GFLOPs. This experiment further demonstrates that JL-guided configurations strike an optimal balance, maintaining competitive performance while ensuring computational efficiency suitable for clinical deployment.

| Dataset | Configuration | MParams ↓ | GFLOPs ↓ | Dice ↑ |
|---|---|---|---|---|
| | $\langle 1, 1, 1, 1\rangle$ | 0.618 -0.091 | 2.637 -0.075 | 37.95 -6.25 |
| Hecktor2022 | $\langle 4, 8, 8, 16\rangle$ | 0.709 | 2.712 | 44.20 |
| | $\langle 16, 32, 64, 128\rangle$ | 1.342 +0.633 | 3.029 +0.317 | 43.21 -0.99 |
| | $\langle 1, 1, 1, 1\rangle$ | 0.629 -0.091 | 2.377 -0.063 | 85.82 -1.00 |
| BraTS2021 | $\langle 4, 8, 8, 16\rangle$ | 0.720 | 2.440 | 86.82 |
| | $\langle 16, 32, 64, 128\rangle$ | 1.353 +0.633 | 2.708 +0.268 | 87.98 +1.16 |

Table 14: Performance comparison of different group size configurations across datasets. The JL-guided configuration $\langle 4, 8, 8, 16\rangle$ is used as the reference baseline.

| Methods | MParams | GFLOPs | Tr. Mem. | Inf. Mem. | Thr. GPU | Dice |
|---|---|---|---|---|---|---|
| Window | 1.51 -0.10 | 2.80 -0.04 | 678 -46 | 1066 +918 | 227.28 +45.83 | 56.01 -5.42 |
| Downsample | 1.52 -0.09 | 2.78 -0.06 | 1066 +342 | 136 -12 | 239.01 +57.56 | 55.18 -6.25 |
| PWA | 1.61 | 2.84 | 724 | 148 | 181.45 | 61.43 |

Table 15: Computation consumption of different attention variants.

## P    COMPARISON OF DIFFERENT ATTENTION MECHANISMS

Under similar computational cost constraints, we replaced PWA with other attention mechanisms, such as window-based multimodal attention and downsampling-based multimodal attention, as shown in Figure 15. Although our method is not optimal due to the larger tensor size change rate, the model performance is significantly better than the other two attention mechanisms, which further validates the effectiveness of PWA for heterogeneous modality modeling.

## Q    COMPARISON OF DIFFERENT KNOWLEDGE TRANSFER STRATEGIES

The comparison results with other strategies are listed in Table 4, with settings largely consistent with the dual-stream settings in reference (Qiu et al., 2023). All comparison methods were performed under the same conditions. The additional training overhead is listed in Table 16, where SDKT uses only about 100 MB more memory than the baseline methods.

## R    SCALING LAW OF VELOXSEG

Our specific parameter configuration is as follows:

- S represents changing the convolution kernel from [1,3,5] to [3] under the original parameter configuration.
- B represents the original parameter configuration.
- B+ represents scaling up the number of attention and convolution channels from 16 to 32.
- L represents scaling up the depth of attention and convolution from 1 to 2.

In VeloxSeg, the Dice value is directly proportional to the number of parameters/floating-point operations: for versions S to B, the Dice value increases by 1.5 for every 0.47M additional parameters; for versions B to B++, the Dice value increases by 1.5 for every 360M additional parameters; while for version L, with fewer parameters/floating-point operations than B++, the Dice value only increases by 0.55. This indicates diminishing returns and that architectural adjustments (not just scaling) are key to improving performance. Non-monotonic resource ordering (L version is smaller than B++ version) results in roughly equal Dice values.

| Methods | MParams | GFLOPs | Tr. Mem. | Inf. Mem. | Thr. GPU | Dice |
|---|---|---|---|---|---|---|
| w/o Teacher | 1.61 | 2.84 | 724 | 148 | 181.45 | 61.43 |
| w Teacher | 1.66 +0.05 | 1.79 -1.05 | 824 +100 | 152 +4 | 390.91 +209.46 | 59.71 -1.72 |
| + $\ell_1$ | 1.66 +0.05 | 1.79 -1.05 | 824 +100 | 152 +4 | 390.91 +209.46 | 1.67 -59.76 |
| + Affinity | 1.66 +0.05 | 1.79 -1.05 | 894 +170 | 152 +4 | 390.91 +209.46 | 41.44 -19.99 |
| + Shared ROI | 1.66 +0.05 | 1.79 -1.05 | 1064 +340 | 152 +4 | 390.91 +209.46 | 57.15 -4.28 |
| + SDKT | 1.66 +0.05 | 1.79 -1.05 | 842 +118 | 152 +4 | 390.91 +209.46 | 62.51 +1.08 |

Table 16: Computation consumption of different knowledge transfer methods.

| Model | Dice | Parameters (M) | FLOPS (G) |
|---|---|---|---|
| nnUNet | 55.85 | 88.62 | 3078.83 |
| VeloxSeg S | 68.56 | 1.19 | 1.41 |
| VeloxSeg B | 70.05 | 1.66 | 1.79 |
| VeloxSeg B+ | 71.56 | 5.26 | 4.27 |
| VeloxSeg L | 72.11 | 2.65 | 2.45 |

Table 17: Accuracy results of VeloxSeg after increasing model size.

