# OpenReview forum: "Johnson-Lindenstrauss Lemma Guided Network for Efficient 3D Medical Segmentation"
_ICLR.cc/2026/Conference — ICLR 2026 Poster_

### Official Review · Reviewer_PE39 · 2025-10-29

**Soundness:** 2
**Presentation:** 3
**Contribution:** 2
**Rating:** 2
**Confidence:** 3

**Summary:**

The paper introduces VeloxSeg, a lightweight 3D medical segmentation model that aims to keep robustness while cutting compute. It uses Paired Window Attention to capture global context with near-linear cost, a Johnson–Lindenstrauss–guided rule to set grouped-convolution sizes so features preserve distance structure, and Spatially Decoupled Knowledge Transfer with Gram-matrix supervision to inject texture priors without extra inference cost. Across some datasets, it reports competitive Dice with markedly higher GPU/CPU throughput than common baselines.

**Strengths:**

* Creative combo of (i) Paired Window Attention for global–local context with near-linear cost, (ii) a Johnson–Lindenstrauss–guided rule to size grouped convolutions, and (iii) Gram-matrix (SDKT) supervision that adds no inference overhead. The JL connection to architectural sizing is a fresh angle for lightweight 3D segmentation.

* Solid empirical scope across multiple 3D medical benchmarks with both accuracy and throughput (GPU/CPU) reported alongside params and GFLOPs. Comparisons include lightweight and multimodal baselines, giving a practical sense of trade-offs.

**Weaknesses:**

* **Inconsistent superiority across metrics.** The model wins on some averages but lags on others (e.g., HD95/precision/recall) with non-trivial gaps.

* **Comparison fairness is under-specified.** Throughput (GPU/CPU) and accuracy comparisons may be skewed by different patch sizes, AMP/FP32, dataloader workers, I/O, pre/post-processing, and TTA.

* **Statistical robustness is weak.** Results appear single-seed without CIs.

* **SDKT attribution unclear.** Teacher details and potential data leakage aren’t fully specified; gains could stem from extra supervision rather than architecture.


* **External validity.** Limited evidence under domain shift (site/scanner).

**Questions:**

* What is the primary target metric (e.g., Dice vs. HD95), and do your claims hold on that metric with mean ± std over ≥3 seeds? Several baselines beat VeloxSeg on non-primary metrics—please provide confidence intervals, exact p-values, and a Pareto plot to make the trade-off explicit.

* Were patch size, AMP/FP32 policy, I/O, dataloader workers, caching, pre/post-processing, and TTA identical across methods? Please re-run baselines in the same codebase and report end-to-end latency on the same hardware, with CPU thread counts fixed.

* All headline results appear single-seed. Can you report 3–5 seeds for every table, include Box/Violin plots, and discuss sensitivity to random init and data order?

* What are the teacher architectures, pretext tasks, and training data? Confirm there is no split leakage. Report the extra training compute and quantify SDKT’s incremental gain vs. strong alternatives under matched budgets.

* At fixed GFLOPs/latency, do PWA and JL still win? Please include budgeted comparisons and scaling curves showing how accuracy changes as compute increases.


**I am open to changing my score based on the author's responses.**

---

> ### Author Response · Authors · 2025-11-22
>
> Thank you for your time with our work. All your concerns are carefully addressed as follows, where Weaknesses and Questions are denoted as **W** and **Q**. We sincerely hope the revised paper by taken into these valuable comments would meet the expectations of the reviewer, and we are glad to provide further responses if needed.
>
> ### W1 & Q1: Primary metric and reasons for lagging behind on some other metrics
>
> **Our primary metric is the Dice coefficient.** We additionally use recall and precision to better understand performance on hard cases. HD95 is reported for completeness but plays a secondary role:
>
> - *HD95 is insensitive to volume coverage and can encourage overly aggressive predictions.*  For datasets with widely distributed lesions, a model that labels almost all suspicious regions as lesions may achieve low HD95, even if the Dice and recall are poor.
> - *The current evaluation protocol tends to ignore hard samples in HD95.* When a small lesion is present (\(N \neq 0\)) but the model completely misses it (predicting no lesion), HD95 becomes NaN and is **excluded** from the average. These are exactly the hardest cases, so excluding them paradoxically leads to **lower average HD95** for conservative models.
>
> ### W2 & Q2: The fairness of the throughput comparison is not clarified
>
> Our latency measurements are obtained using a **unified benchmarking script** adapted from EfficientViT [1], which is specifically designed to ensure fair comparisons. We also provide our code link in Appendix A. We control for the following:
>
> - identical patch size,
> - FP32 precision (no AMP),
> - random synthetic input (to ensure identical I/O constraints),
> - identical dataloader settings (number of workers, caching),
> - identical preprocessing/postprocessing,
> - No TTA for latency evaluation.
> - No hardware acceleration trick.
>
> As explicitly stated in Figure 1, all latency measurements are performed on a **single RTX 3090 GPU** and a **single CPU thread**.
>
> ### W3 & Q1 & Q3: Multiple random seeds
>
> We appreciate your concern about randomness. Different seeds affect parameter initialization and data ordering, which can influence final performance. To quantify this effect, we **retrain the SOTA model from each family with 4 different random seeds** and report the key results. We additionally provide **boxplot/violin-plot visualizations** in Appendix T. These experiments show that the **variance across seeds is small**, and our main results are statistically stable under the standard deviations observed.
>
> ### W4 & Q4: Concerns about data breaches in SDKT and a comparison with other strategies
>
> **There is no data leakage in SDKT.** The teacher branch is **not pre-trained** on any external task. Instead, it is trained **jointly with the main network** in an end-to-end manner, following the training framework of [2]. To avoid ambiguity, we revise the SDKT section to explicitly present the **final loss function**.
>
> **The comparisons with other strategies are listed in Table 4**, with settings largely consistent with the dual-stream setup in reference [2]. All comparison methods are under the same conditions.
>
> **The additional training overhead is listed in Table 16**, where SDKT uses only approximately 100 MB more memory than the baseline methods. Furthermore, we *additionally report the GPU memory usage comparisons of our method with other methods*, as shown in Table 8 and Figure 9.

---

> > ### Author Response · Authors · 2025-11-22
> >
> > ### W5: Limited evidence under domain shift (site/scanner)
> >
> > **Employment of Multicenter Datasets**. Our datasets are all multicenter datasets, each aggregated from multiple sites and scanners.
> >
> > ### Q5: Comparison of attention and convolution at similar costs, and accuracy-computation cost scaling curves
> >
> > **Experimental study of replacing PWA with "window/downsampling attention" (multi-modal)**. The results are shown in Table 15.
> >
> > | Methods | MParams | GFLOPs | Train Mem. | Inference Mem. | Thr. GPU | Dice |
> > | :--- | :---: | :---: | :---: | :---: | :---: | :---: |
> > | Window | 1.51 | 2.80 | 678 | 1066 | 227.28 | 56.01 |
> > | Downsample | 1.52 | 2.78 | 1066 | 136 | 239.01 | 55.18 |
> > | PWA | 1.61 | 2.84 | 724 | 148 | 181.45 | 61.43 |
> >
> > Comparison of convolution accuracy at similar costs is given in Figure 6.
> >
> > Based on your suggestion, we appropriately scaled our model:
> >
> > - S represents changing the convolution kernel from [1,3,5] to [3] under the original parameter configuration
> > - B represents the original parameter configuration
> > - B+ represents scaling up the number of attention and convolution channels from 16 to 32
> > - L represents scaling up the depth of attention and convolution from 1 to 2
> >
> > Based on this, we summarize the accuracy and corresponding computational cost of **training S/B/B+/L on AutoPETII using nnUNet** as follows:
> >
> > | Model | Dice | Parameters (M) | FLOPS (G) |
> > | :--- | :--- | :--- | :--- |
> > | **VeloxSeg Small** | 68.56 | 1.19 | 1.41 |
> > | **VeloxSeg Base** | 70.05 | 1.66 | 1.79 |
> > | **VeloxSeg Base+** | 71.56 | 5.26 | 4.27 |
> > | **VeloxSeg Large** | 72.11 | 2.65 | 2.45 |
> >
> > ### R4 References
> >
> > [1] Liu X, Peng H, Zheng N, et al. "EfficientViT: Memory efficient vision transformer with cascaded group attention." CVPR, 2023.
> > [2] Qiu Z, Hu Y, Chen X, et al. "Rethinking dual-stream super-resolution semantic learning in medical image segmentation." TPAMI, 2023.

---

### Official Review · Reviewer_L51F · 2025-10-31

**Soundness:** 3
**Presentation:** 3
**Contribution:** 3
**Rating:** 6
**Confidence:** 3

**Summary:**

The paper introduces VeloxSeg, a lightweight and theoretically grounded 3D medical segmentation framework designed to resolve the efficiency–robustness conflict in medical imaging. It integrates: Paired Window Attention (PWA) for efficient multi-scale context capture, Johnson–Lindenstrauss lemma-guided convolution (JLC) to preserve spatial adjacency while reducing parameters, and Spatially Decoupled Knowledge Transfer (SDKT) using Gram matrices for knowledge distillation without inference overhead. Evaluations on four multimodal datasets (AutoPET-II, Hecktor2022, BraTS2021, MSD2019) show superior Dice scores and computational efficiency compared to state-of-the-art models.

**Strengths:**

- Evaluations span multiple datasets and architectures (CNN, Transformer, KAN, RWKV, and Mamba hybrids). The paper includes comprehensive ablation studies, efficiency tests (GPU/CPU throughput), and shows consistent performance gains.

- Strong theoretical grounding via the Johnson–Lindenstrauss lemma. The proposed JLC introduces a sound theoretical perspective into the design of lightweight segmentation networks.

**Weaknesses:**

- In medical image segmentation tasks, using a fixed train/validation/test split (e.g., 6:2:2) can lead to unstable or biased results, particularly due to typically small dataset sizes and high inter-sample variability. A more robust and widely accepted approach in this domain is k-fold cross-validation, which helps assess the model’s generalizability more reliably. It is unclear whether the evaluation results reported in Table 1 are obtained using multiple folds or a single split. If only a single split is used, the reported performance may not be representative and could be sensitive to the particular partitioning. I recommend clarifying this point and, if not already done, performing evaluation using multiple folds or reporting variance across different splits.

- I noticed that the baseline model (nnU-Net) is, by default, trained for 1000 epochs on the AutoPET-II dataset, whereas in your paper, all models are trained for only 300 epochs. This raises concerns about whether all models were evaluated at convergence and whether the comparisons reflect fully optimized performance for each method.

- The AMOS dataset is currently one of the most widely adopted benchmarks for large-scale 3D medical image segmentation, and it also includes multimodal data (CT and MRI). It is unclear why your experiments do not include evaluation on AMOS. I recommend that the authors justify their dataset choices and discuss whether the proposed method generalizes well to other benchmarks.

**Questions:**

See the Weaknesses. I am open to discussing this further during the rebuttal and will be happy to increase my score if my concerns are addressed.

---

> ### Author Response · Authors · 2025-11-22
>
> Thank you for your time with our work. All your concerns are carefully addressed as follows, where Weaknesses and Questions are denoted as **W** and **Q**. We sincerely hope the revised paper by taken into these valuable comments would meet the expectations of the reviewer, and we are glad to provide further responses if needed.
>
> ### W1: K-fold cross-validation to demonstrate robustness to data splits
>
> Our main results are based on **single-fold cross-validation** experiments. Based on your suggestion, we perform **4-fold cross-validation** on the AutoPET-II and Hecktor2022 datasets for the **SOTA in each model type** to verify the representativeness and robustness of these results.
>
> | Model         | Metric    | Mean (%) | Std (%) | 95% CI (%) |
> |---------------|-----------|----------|---------|------------|
> | VeloxSeg      | Dice      | 62.31    | 1.32    | 2.10       |
> |               | Precision | 64.14    | 2.63    | 4.18       |
> |               | Recall    | 69.87    | 2.72    | 4.33       |
> | VSmTrans      | Dice      | 60.65    | 2.18    | 3.47       |
> |               | Precision | 70.43    | 3.91    | 6.23       |
> |               | Recall    | 61.08    | 7.26    | 11.56      |
> | U-RWKV        | Dice      | 58.19    | 1.04    | 1.65       |
> |               | Precision | 62.77    | 3.86    | 6.14       |
> |               | Recall    | 66.91    | 6.68    | 10.63      |
> | H-Denseformer | Dice      | 59.65    | 2.46    | 3.92       |
> |               | Precision | 66.04    | 3.61    | 5.75       |
> |               | Recall    | 64.16    | 10.38   | 16.52      |
>
> As shown in the table aboves, model performance is **insensitive to the specific data partitioning**, and the main metric we report is representative.
>
> ### W2: Concern about the number of training epochs
>
> We set the number of epochs to **300**, which is a common choice in recent literature. It is important to note that the "1000 epochs" in nnU-Net are not directly comparable: nnU-Net uses **250 iterations per epoch**, for a total of \(250 \times 1000 = 250{,}000\) iterations. In our setting, each epoch corresponds to a **full pass over the dataset**. Taking into account that we use 60% of the training data per epoch (random sampling), the number of iterations is:
>
> - AutoPET-II: \(300 \times 1014 \times 0.6 = 182{,}520\),
> - BraTS2021: \(300 \times 1251 \times 0.6 = 225{,}180\).
>
> Thus, our total number of iterations is **of the same order** as nnU-Net, and adapting the iteration count to the dataset size is natural. We also provide examples of epoch settings in related works:  TransUNet uses 150 epochs [1], U-KAN uses 400 [2], U-RWKV uses 200 [3], NestedFormer uses 300 [4], and H-DenseFormer uses 100 [5].
>
> **Convergence:** To directly address your concern, we further plot the **training loss curves** of VSmTrans, the model with the largest number of parameters. As shown in Figure 12, the model has fully converged by the end of training.

---

> > ### Author Response · Authors · 2025-11-22
> >
> > ### W3: Clarifying the rationale behind dataset selection
> >
> > **Reasons for not using Amos as the primary dataset** While Amos is indeed a widely used benchmark dataset for 3D medical segmentation, its primary design purpose is to evaluate **modal generalization ability**: CT and MR are not paired, and the model needs to independently accept inputs from both modalities [6]. Our work focuses on **heterogeneous modalities and multimodal fusion**, where multiple modalities are input simultaneously for the same case, requiring the model to explore complementary information between them. This is fundamentally different from Amos's modal generalization task.
> >
> > Running our method on Amos actually shifts from **multimodal-specific modeling** to **early fusion**, which **weakens our approach**. Nevertheless, we are pleased to present a comparison of our method with nnUNet on Amos, as shown below:
> >
> > | Model | Dice | Parameters (M) | FLOPS (G) |
> > | :--- | :--- | :--- | :--- |
> > | **VeloxSeg S** | 79.95 | 1.31 | 7.81 |
> > | **VeloxSeg B** | 81.81 | 1.78 | 8.25 |
> > | **VeloxSeg L** | 83.97 | 2.62 | 8.95 |
> > | **VeloxSeg Base+** | 84.90 | 5.01 | 17.16 |
> > | **nnUNet** | 87.78 | 88.42 | 4424.38 |
> >
> > - S represents changing the convolution kernel from [1,3,5] to [3] under the original parameter configuration.
> > - B represents the original parameter configuration.
> > - B+ represents scaling up the number of attention and convolution channels from 16 to 32.
> > - L represents scaling up the depth of attention and convolution from 1 to 2.
> >
> > **Dataset selection follows two criteria**:
> >
> > - **Cross-region coverage.** We use datasets where lesions are distributed across different body regions (not just a single organ).
> > - **High modal heterogeneity.** We specifically choose datasets with high modal heterogeneity (e.g., PET vs. CT) to test the performance of multimodal fusion.
> >
> > These factors significantly increase the segmentation difficulty and can expose potential flaws in lightweight architectures. Ablation experiments conducted under such challenging conditions yielded configurations that were **more easily generalized to** simpler tasks, as evidenced by Tables 1, 3, 10, and 14.
> >
> > ### R3 References
> >
> > [1] Chen J, Lu Y, Yu Q, et al. "TransUNet: Transformers make strong encoders for medical image segmentation." arXiv:2102.04306, 2021.
> > [2] Li C, Liu X, Li W, et al. "U-KAN makes strong backbone for medical image segmentation and generation." AAAI, 2025.
> > [3] Ye H, Tang F, Zhao P, et al. "U-RWKV: Lightweight medical image segmentation with direction-adaptive RWKV." MICCAI, 2025.
> > [4] Xing Z, Yu L, Wan L, et al. "NestedFormer: Nested modality-aware transformer for brain tumor segmentation." MICCAI, 2022.
> > [5] Shi J, Kan H, Ruan S, et al. "H-DenseFormer: An efficient hybrid densely connected transformer for multimodal tumor segmentation." MICCAI, 2023.
> > [6] Ji Y, Bai H, Ge C, et al. "Amos: A large-scale abdominal multi-organ benchmark for versatile medical image segmentation." NeurIPS, 2022.
> > [7] Gatidis S, Hepp T, Früh M, et al. "A whole-body FDG-PET/CT dataset with manually annotated tumor lesions." Scientific Data, 2022.
> > [8] Huang Z, Wang H, Deng Z, et al. "Stu-Net: Scalable and transferable medical image segmentation models empowered by large-scale supervised pre-training." arXiv:2304.06716, 2023.

---

### Official Review · Reviewer_syAZ · 2025-11-01

**Soundness:** 3
**Presentation:** 3
**Contribution:** 3
**Rating:** 6
**Confidence:** 4

**Summary:**

This paper proposes an efficient 3D medical image segmentation framework called VeloxSeg. The framework consists of three key components working together: 1) a theory-driven group convolution (JLC) inspired by the Johnson-Lindenstrauss (JL) lemma, serving as the encoder backbone to efficiently compress and encode features; 2) a progressive window attention (PWA) mechanism in the decoder to capture multi-scale contextual information; and 3) a size-aware dynamic knowledge transfer strategy (SDKT) to improve the model's segmentation performance for lesions of different sizes through knowledge distillation. Experimental results on public datasets such as AutoPET-II and Hecktor2022 demonstrate that the proposed method achieves competitive segmentation performance while maintaining high efficiency.

**Strengths:**

- Theoretical grounding for depthwise separable convolution: The use of the JL lemma to derive a lower bound on convolution group size is a notable attempt to bridge architectural design with theoretical guarantees, particularly in preserving adjacency relationships within high-dimensional feature space.

- Comprehensive ablation and visualization analysis: The paper provides detailed visualizations that help validate the contribution of each proposed component, enhancing the overall interpretability and credibility of the approach.

- Good performance-efficiency balance: The most significant advantage of this paper is that it achieves state-of-the-art (SOTA) or highly competitive performance on challenging 3D medical image segmentation tasks while maintaining low computational costs. This is especially important for resource-constrained scenarios in clinical applications.

**Weaknesses:**

### Mismatch Between Claimed Contributions and Experimental Evidence

The paper’s title and introduction highlight **JLC** as the core and primary contribution. However, based on the ablation study (Table 2), the impact of the JLC module does not appear to be the most significant. For instance, adding the PWA module on top of the JLC encoder brings a substantial performance boost (Dice score increases from around 55% to 61%). This suggests that the model’s success heavily relies on the strong capabilities of **PWA** and **SDKT**, as well as their synergistic interaction, rather than solely on the superiority of the JLC encoder. The current narrative may overemphasize the dominance of JLC. It is recommended that the authors reconsider and more accurately position the core contribution of the paper. Perhaps as the design of an efficient and cohesive system, rather than a single JLC module.

---

### Theoretical vs Empirical Gap

While the theoretical motivation behind JLC is a notable strength, the leap from the theoretical boundary
$ C_{group} \geq c_{JL} \varepsilon^{-2} \log N(M, v) $
to the empirical approximation $ N(M, v) \approx (M \cdot v)^{\alpha} $ is substantial. The paper acknowledges this, but in practice, the final group size configuration (e.g., {n, 2n, 2n, 4n}) is still determined empirically by tuning α or n. This weakens the claim that the design is “fully theory-driven.” It might be more accurate to describe it as theory-inspired or theory-motivated.

---

### System Complexity and Research Depth

VeloxSeg integrates three independent and non-trivial modules, making it a complex system. Although the ablation results demonstrate overall effectiveness, such complexity makes it difficult to fully disentangle and analyze the individual contributions of each module. For a venue like ICLR, which often values deep exploration of a single core idea, a paper that focuses exclusively on JLC—examining its representational properties, limitations, and task adaptability in depth—might be equally valuable. From a practical standpoint, this complexity could also pose challenges for researchers or practitioners attempting to adapt and fine-tune the model.

---

### Hyperparameter Sensitivity of the PWA Module

The design of the PWA module, particularly the choices of minimum/maximum window sizes and dilation rates, appears to be dataset-dependent (Appendix D.1 notes different setups for AutoPET-II and Hecktor2022). While such dependency is common in deep learning models, providing a sensitivity analysis on these hyperparameters and offering general configuration guidelines would greatly enhance the reproducibility and generalizability of the proposed method.

---

### Others

- PWA: Choosing and aligning paired small and large windows is nontrivial, especially for 3D data. In real-world scenarios where modalities (e.g., PET/CT or MRI/CT) have different voxel spacings or mis-alignment, the spatial correspondence between large and small windows can become ambiguous. It remains unclear whether the proposed spatial window mechanism can effectively handle such cases.

- Long-range understanding: The long-range dependency modeling capability of PWA should be further justified. Including attention visualizations would strengthen the understanding of how multi-modal information is integrated through attention mechanisms.

- Depthwise convolution and adjacency: In Section 2.3, the authors state that “depthwise convolution destroys the adjacency relationship between data in the feature space” (Figure 3(b)). However, it is unclear whether this claim holds true in practice. Providing theoretical justification or visual evidence would help substantiate this assertion.

- Efficiency and SDKT: The teacher training cost of the Spatially Decoupled Knowledge Transfer (SDKT) module remain concerns, as it requires an additional training phase for the teacher network.

**Questions:**

### On the Design of JLC

Could you elaborate on the reasoning behind choosing the approximation
$ N(M, v) \approx (M \cdot v)^{\alpha} $?
Did you explore alternative functional forms or methods to estimate the complexity of the data manifold?
How sensitive is the final model’s performance to the scaling factor **α** (or the base channel number n)?
A more detailed sensitivity analysis would greatly help readers understand the robustness of the JLC design.

---

### Further Clarification on JLC’s Contribution

To better distinguish the unique theoretical contribution of JLC, did you attempt to combine a more naive lightweight convolutional module, such as a standard group convolution with empirically chosen group sizes, or a depthwise separable convolution, with PWA and SDKT?
This comparative experiment is crucial. It helps determine whether the key advantage truly lies in JLC’s theory-guided design principle itself, or whether any reasonably designed lightweight encoder could achieve comparable performance when integrated with strong modules like PWA and SDKT.

---

Others please see the Weaknesses section for details.

---

> ### Author Response · Authors · 2025-11-22
>
> Thank you for your time with our work. All your concerns are carefully addressed as follows, where Weaknesses and Questions are denoted as **W** and **Q**. We sincerely hope the revised paper by taken into these valuable comments would meet the expectations of the reviewer, and we are glad to provide further responses if needed.
>
> ### W1 & Q2: Given that other modules contributed significantly more performance improvements, the core role of JLC has been questioned. Results from depth-waise convolution + PWA + SDKT need to be added
>
> JLC is the core component of the framework, ensuring our model is easy to train, has high throughput, and stable representations.
>
> The significant performance improvements achieved by PWA and SDKT are due to their robust representation based on JLC. We supplement this with experiments using Depthwise Convolution + PWA + SDKT, as shown below:
>
> | Method               | Dice  |
> |----------------------|-------|
> | DWC (kernel=<1,3,5>) | 53.65 |
> | DWC+PWA+SDKT         | 57.55 |
> | JLC (kernel=<1,3,5>) | 55.84 |
> | JLC+PWA+SDKT         | 62.51 |
>
> With additional knowledge injection, the accuracy gap between DWC and JLC widens from 2.19% to 4.96%. This is because DWC cannot preserve the original relationships between tokens (already modeled features). This further illustrates the importance of **JLC as a core component** of the model backbone, serving as the **foundation for the effectiveness of other modules**.
>
> ### W2: Overstated claim of being "fully theory-driven"
>
> In the paper, we consistently describe JLC as **"theory-guided"**, not "fully theory-driven". We therefore believe this particular weakness does not apply.
>
> "Theory-guided" not only reflects that we are inspired by theories, but also emphasizes that we conduct further empirical research based on these theories. This aligns better with our technological roadmap than simply being "theory-inspired" or "theory-motivated".
>
> ### W3: Concerns that complex systems may be difficult for practitioners to use, and suggestions for further in-depth research on JLC
>
> We fully share your concern about usability. **VeloxSeg exposes only three hyperparameters to end users: `in_ch`, `base_ch`, and `depth`.** The effective number of tunable knobs is comparable to many popular baselines.
>
> - **`in_ch` (a list):** This controls how the Transformer encoder handles modality-specific vs. mixed-modality modeling. The number of Transformer branches equals the length of `in_ch`, and the sum of its elements equals the number of input modalities. For example, with two input modalities, `in_ch` can be `[1, 1]` or `[2]`.
>   - `[1, 1]`: two Transformer encoders, each processing one modality separately (**modality-specific modeling**).
>   - `[2]`: one Transformer encoder processing the two modalities jointly (**early fusion**).
>   - We recommend modality-specific modeling (e.g., `[1, 1]`) when modalities are highly heterogeneous, and mixed modeling (e.g., `[2]`) otherwise to save computation.
> - **`base_ch` and `depth`:** If computational resources allow, we recommend moderately increasing `base_ch` and `depth`. To facilitate this, we provide three reference configurations (S / B / L) in the experiments.
>
> **The current approach, based on theoretical guidance and experimental verification, is both efficient and rigorous for developing sufficiently lightweight, fast, and robust model architectures**. If JLC were based on more mathematical assumptions, it could obtain tighter upper bounds on the manifold coverage number for a single dataset. However, due to the scaling of logarithms in the JL lemma and the fact that group size is most importantly divisible by the number of input channels of the convolution, insufficient perturbations will not affect the final group size configuration; see Reviewer syAZ's response to Question 1 for details.

---

> > ### Author Response · Authors · 2025-11-22
> >
> > ### W4: Sensitivity of PWA to hyperparameters
> >
> > Although PWA appears to involve multiple parameters, they are in fact **not free hyperparameters**. They are uniquely determined by three design principles (explained in detail in the appendix):
> >
> >   1. *Cover the global spatial range*.
> >   2. *Ensure that the number of windows evenly divides the channels and is as large as possible*.
> >   3. *Enable fine-grained modeling*.
> >
> > Once the **`input size`** and **`base_ch`** are given, these principles determine a **unique** set of paired windows. For example, on AutoPET-II, the input image patch size is \((96, 96, 96)\). After image patch embedding, the first-stage feature map size is \((24, 24, 24)\), with 16 channels. There is **exactly one valid configuration**:
> >
> > |                  | 1         |     2     | 3            |4             |
> > |------------------|-----------|-----------|--------------|--------------|
> > | Big window set   | [3, 3, 3] | [6, 6, 6] | [12, 12, 12] | [24, 24, 24] |
> > | Small window set | [1, 1, 1] | [2, 2, 2] | [4, 4, 4]    | [8, 8, 8]    |
> > | Expansion ratio  | 2         |           |              |              |
> >
> > The constraints are:
> >
> > 1. The final big-window size must reach `[24, 24, 24]`;
> > 2. The initial small-window size must start from `[1, 1, 1]`;
> > 3. The expansion ratio must be the minimal value 2, and each paired window must expand until further expansion is impossible. This is because PWA is designed **not only to reduce computation**, but also to **reduce redundancy across attention heads** (as illustrated by the differences between Figure 4(a) and 4(b),(c)).
> >
> > Therefore, we provide a **recommended general configuration** that only depends on the image size, similar to other Transformer-based methods such as UNETR [1] and Swin UNETR [2]:
> >
> > | Parameter                | Stage | Value               |
> > |--------------------------|-------|---------------------|
> > | min_big_window_sizes     | 1     | [H/32, W/32, D/32]  |
> > |                          | 2     | [H/16, W/16, D/16]  |
> > |                          | 3     | [H/32, W/32, D/32]  |
> > |                          | 4     | [H/32, W/32, D/32]  |
> > | min_small_window_sizes   | 1     | [1, 1, 1]           |
> > |                          | 2     | [1, 1, 1]           |
> > |                          | 3     | [1, 1, 1]           |
> > |                          | 4     | [1, 1, 1]           |
> > | r (expansion ratio)      | ALL     | 2                 |
> >
> > ### W5: Concerns about 3D registration and spatial misalignment
> >
> > In the preprocessing stage, all images are resampled to a **common voxel spacing/registration space**. Without this step, none of the models (including baselines) could handle different input resolutions consistently.
> >
> > In addition, PWA uses **relative positional encoding** based on physical spacing and operates at the **window level** rather than requiring voxel-wise strict alignment. This design makes PWA relatively robust to small residual misalignments. More detailed preprocessing information is provided in Appendix G.
> >
> > ### W6: Validation of PWA’s long-range modeling capability and feature visualization
> >
> > Figure 4 reports the **mean attention distance (MAD)** to demonstrate that PWA effectively captures **cross-scale and long-range dependencies**. The exact calculation formula is given in Appendix N, following the definition in ViT [4].
> >
> > Figure 5 shows the visualization results of attention-related features to intuitively demonstrate the feature changes after PWA modeling multimodalities. We also provide further literature [5,6,7] that illustrates how multimodal information interacts through attention mechanisms.
> >
> > ### W7: Visual evidence that depth-wise convolutions destroy adjacency in the feature space
> >
> > We agree that this needs to be supported by visualization. Our argument is based on:
> >
> > - **Several empirical phenomena** discussed in our responses to Reviewer kwQp's weaknesses 3 and questions 2;
> > - **t-SNE visualizations** of feature embeddings for depthwise convolution and JLC, as shown in Figure 11. This provides more direct evidence for our argument, showing that JLC ​​preserves adjacency relationships better than depthwise convolution in the learned feature space.

---

> > > ### Author Response · Authors · 2025-11-22
> > >
> > > ### W8: Concern about the extra training cost and potential additional training stage for SDKT
> > >
> > > **SDKT does not require an extra training stage.** It is activated **synchronously** during the training of the main backbone, following the dual-stream training strategy proposed in [3].
> > >
> > > **The additional training cost of SDKT is low.** While your concern about extra trainable branches is valid, our base architecture is lightweight enough that SDKT adds only about **100 MB of GPU memory** during training, as reported in Table 15.
> > >
> > > In addition, SDKT achieves **higher gains** than alternative designs (L1, Affinity, Shared-ROI) **under the same or even lower training budget**, which further justifies its efficiency.
> > >
> > > ### Q1: Further explanation of JLC and why we choose this particular approximation and attempts with other approximation functions
> > >
> > > **Our function motivated from [8] (Definition 10.15 and Exercise 10.17)**. The covering number is bounded by $N \le C(1/\epsilon)^\beta$, where $C \ge 1$ is a constant independent of the ball, $\epsilon\in (0,1)$. To ensure generality, we do not make further assumptions about the data to obtain specific parameters, but instead use empirical functions for approximation: **$\hat{N} = (M \cdot v)^\alpha$** . In this formulation:
> > >
> > > - **$M \cdot v$** replaces $1/\epsilon$, representing the *coverage density* required per dimension. Given the constraints of JL's logarithmic scaling and the requirement that the group size divides the input channel, we omit the possible constant term.
> > > - **$\alpha$** serves as a *difficulty coefficient* reflecting the dataset's intrinsic complexity. We calibrate $\alpha$ based on the most challenging dataset to ensure robust generalization across different tasks.
> > >
> > > **More fine-grained theoretical modeling brings limited practical benefit.** As detailed in R2 Q1, small numerical perturbations do not change the discrete choice of group size, because the group size must divide the base number of channels. Therefore, the **theory-guided + empirical** strategy is both **rigorous and efficient** for our purpose.
> > >
> > > **Unnecessary to extend the approximation to more complex polynomials.** The covering-number estimate is only an intermediate step; JL lemma then applies a logarithm. Consequently, low-degree and constant terms in a polynomial approximation have a **minimal effect** after taking the log. For example, consider \(v = 43, M = 2, \alpha = 1\). If we add a constant term \(\beta\), $\log(Mv + \beta) - \log(Mv) = \log(1 + \beta / 128)\approx 0$. Besides, because the `group_size` must divide `input_channel`, such a small interrupt rarely changes the final group size.
> > >
> > > **Our robustness stems from 3 points:**
> > >
> > > - The logarithm in JL lemma reduces parameter perturbation.
> > > - The constraint that the group size must be divisible by the number of input channels ensures that minor perturbations cannot lead to configuration changes.
> > > - Parameters determined on a harder dataset generalize well. AutoPETII is a whole-body PET/CT tumor segmentation dataset with: 1) heterogeneous modalities (PET vs. CT); 2) tumors distributed throughout the body; 3) extremely large target size variation (\(10 \sim 10^5\)); 4) "camouflaged" tumor segmentation.
> > >
> > > We therefore **fit \(\alpha\) using AutoPETII**, and then reuse it on Hecktor2022 and BraTS2021. Our robustness can be seen from Tables 1, 3, 10, and 14.
> > >
> > > ### R2 References
> > >
> > > [1] Ali H., Yucheng T., Vishwesh N., et al. "UNETR: Transformers for 3D Medical Image Segmentation." WACV, 2022.
> > > [2] Ali H., Vishwesh N., Yucheng T., et al. "Swin UNETR: Swin Transformers for Semantic Segmentation of Brain Tumors in MRI Images." MICCAI, 2021.
> > > [3] Qiu Z, Hu Y, Chen X, et al. "Rethinking dual-stream super-resolution semantic learning in medical image segmentation." TPAMI, 2023.
> > > [4] Alexey D., et al. "An Image is Worth 16×16 Words: Transformers for Image Recognition at Scale." ICLR, 2021.
> > > [5] Shi J, Kan H, Ruan S, et al. "H-denseformer: An efficient hybrid densely connected transformer for multimodal tumor segmentation." MICCAI, 2023.
> > > [6] Xing Z, Yu L, Wan L, et al. "NestedFormer: Nested modality-aware transformer for brain tumor segmentation." MICCAI, 2022.
> > > [7] Lu J, Chen J, Cai L, et al. H2ASeg: Hierarchical adaptive interaction and weighting network for tumor segmentation in PET/CT images. MICCAI, 2024.
> > > [8] Heinonen, J. "Lectures on Analysis on Metric Spaces". Springer-Verlag, 2001.

---

> ### Comment · Reviewer_syAZ · 2025-11-26
> **Reply to rebuttal**
>
> The rebuttal is thorough.
> It addressed good partition of my concerns.
>
> I have increased my score accordingly.
>
> Best,
>
> Reviewer syAZ

---

> ### Author Response · Authors · 2025-11-26
>
> Dear Reviewer syAZ,
>
> Thank you very much for your follow-up and for carefully reading our rebuttal.
>
> We truly appreciate your constructive comments and your decision to increase the score of our paper. If you have any further questions or suggestions, we would be very happy to respond.
>
> Best regards,
>
> Authors of Submission3962

---

### Official Review · Reviewer_kwQp · 2025-11-01

**Soundness:** 3
**Presentation:** 4
**Contribution:** 2
**Rating:** 4
**Confidence:** 4

**Summary:**

The authors propose VeloxSeg, a dual stream convolution and transformed based medical image segmentation method. The authors introduce a Johnson-Lindenstrauss theorem guided setting for grouped convolutions for extracting local features at minimized cost, a Paired Window Attention block for extracting local and global features across modalities and a Spatially Decoupled Knowledge Transfer strategy for learning rich textual features. They demonstrate superior performance against 19 baselines on 2 datasets, alongside ablation experiments and analysis of their model performance.

**Strengths:**

1. The analysis of the authors on their proposed methods and against their 19 baselines is impressive, as is their ablation experiments.
2. The authors use volumetric and surface based metrics to demonstrate the efficacy of their methods.
3. The authors compare against a large number of baselines including categories of models such as multimodal and lightweight methods.
4. The use of Johnson-Lindenstrauss theorem to add grouped convolutions instead of depthwise convolutions is an interesting approach.
5. The SKDT supervision is a novel way to add to their method.

**Weaknesses:**

1. With research papers on advanced architectures, the results are significantly more convincing when compared against equivalent powerful architectures in the field, such as nnUNet ResEncL [4], STU-Net [2], MedNeXt-L [3] which have been compared in public benchmarks as described in [1, 4]. Architectures like MedNeXt are also dependent on depthwise convolutions, which the authors should demonstrate struggle to perform against their JL Convs.

2. The Paired Window Attention is rather difficult to understand in the main text - a bit more explanation would have been useful. This includes while looking at the code in the appendix. The idea of modal features is also somewhat confusing - it is unclear if the authors mean features from different imaging modalities - It should also then be made clear to the readers that their modes range from 2-4 given their choice of datasets.

3. The clever application of the Johnson-Lindenstrauss theorem is appreciated. However, I am not completely convinced of the idea of depthwise convs completely destroying spatial adjacency in feature learning. I agree that they would do so in principle if they were ever applied completely in isolation. But in execution, they are likely followed by a layer of pointwise convs. This is true for some of the authors’ own references - Shufflenet V2 and MedNeXt.

4. I have some concerns regarding the domain generalization experiments by training on BraTS2021 and evaluating on MSD01 Brain Tumor. BraTS2021 reuses data from older BraTS cohorts while MSD01 reuses subsets of BraTS16 and 17. While I could not directly trace sources given the long chain of data reuse in BraTS, there is some concern that both datasets used by the authors might share old cohorts of BraTS 2012 or 2013 data.

References:

[1] Bassi, Pedro RAS, et al. "Touchstone benchmark: Are we on the right way for evaluating ai algorithms for medical segmentation?." NeurIPS, 2024.

[2] Huang, Ziyan, et al. "Stu-net: Scalable and transferable medical image segmentation models empowered by large-scale supervised pre-training." arXiv preprint arXiv:2304.06716 (2023)..

[3] Roy, Saikat, et al. "Mednext: transformer-driven scaling of convnets for medical image segmentation." MICCAI, 2023.

[4] Isensee, Fabian, et al. "nnu-net revisited: A call for rigorous validation in 3d medical image segmentation." MICCAI, 2024.

**Questions:**

1. I recommend that the authors consider restructuring the analogy in lines 72-90 in the introduction. It is rather cumbersome to picture and a simple scientific description would be better. I would also request that they take out references to case, clues, detectives and profilers throughout the text. It is difficult to read in my opinion.

2. Comparisons against architectures such as ResEncL, STUNet and MedNeXt-L are prescient especially given the fact that the paper clearly states that the JL Convolution proposed is designed to counter - as stated by the authors - the phenomenon of “depth-wise convolution destroys the adjacency relationship between data in the feature space” which architectures like MedNeXt are based on. This would also address my concerns in the above weaknesses section regarding depthwise convs.

3. Please investigate my concerns regarding the possible unintentional data leakage in the domain generalization experiments. It is acceptable to simply drop the domain generalization experiment if there is or find a replacement dataset.

4. As the authors state that SKDT helps with rich texture details in feature learning, maybe an idea of HD95 in Table 2 might help their case.

5. The authors need to explain PWA somewhat more in the text in a better way. It is difficult to understand as it currently stands.

---

> ### Author Response · Authors · 2025-11-22
>
> Thank you for your time with our work. All your concerns are carefully addressed as follows, where Weaknesses and Questions are denoted as **W** and **Q**. We sincerely hope the revised paper by taken into these valuable comments would meet the expectations of the reviewer, and we are glad to provide further responses if needed.
>
> ### W1 & Q2: Adding comparisons with ResEncL, STUNet and MedNeXt-L
>
> Based on your thorough consideration of fairness in the nnUNet revised [3], we decide to add MedNeXt [1] as a comparative experiment instead of introducing Stu-Net [4]. The specific reasons are as follows:
>
> - We **add experiments using MedNeXt** in the paper so that readers can more clearly understand the advantages of JLC over depthwise convolutions in a controlled environment.
> - **Using Stu-Net [4] as a baseline is unfair** because it relies on large-scale external supervision and pre-training.
> - **ResEncL [3] is just a UNet with residual connections**. Most of the 17 baseline methods we compared employ residual connections or similar techniques. Therefore, we believe that existing baseline methods are sufficient to demonstrate the effectiveness and competitiveness of our method.
>
> Please see Figure 1, Table 1 and Table 10 for the **new segmentation performance**. You may also be interested in the **peak GPU memory usage** for all methods, as shown in Table 8 and Figure 10.
>
> ### W2 & Q1 & Q5: The PWA analogy in the introduction and JL's concept of modal features may confuse first-time readers
>
> We completely **replace the analogy with a clear scientific description** in the revised version. Thank you for your careful reading; this significantly improves the readability and accuracy of the article.
>
> While the concept of modal features does represent different imaging modalities, **JLC does not require configuration adjustments** for different numbers of modalities. We provide group size settings for 1, 2, and 4 modalities in Appendix E. These values ​​have only a slight impact after the logarithmic operation of the JL lemma, and considering the limitation that group convolution must divide the input channels of the convolution, no specific parameter configuration adjustments are necessary. We provide a more detailed analysis in our response to Reviewer syAZ's Question 1.
>
> ### W3 & Q2: Concern about depth-wise convolutions destroying adjacency, given that MedNeXt and related methods still adopt this design
>
> **We have in fact experimented with MedNeXt-S [1] at an early stage**:
>
> | Metric                     | AutoPETII | Hecktor2022 | BraTS2021 |
> |----------------------------|-----------|-------------|-----------|
> | Dice on Three Datasets (%) | 53.94     | 47.22       | 90.70     |
> | Training Memory (MB)       | 17,216    | 20,372      | 17,258    |
> | Inference Memory (MB)      | 3,678     | 4,376       | 3,734     |
>
> *(Params: 5.54 M, FLOPs: 57.93 G, Throughput GPU: 27.95 patches/s on (96,96,96) patch size)*
> We observe that the performance of MedNeXt-S collapses on AutoPETII and Hecktor2022, and its computational cost is also very high. One plausible reason is that MedNeXt essentially adopts the 2D image-classification hyperparameters from ConvNeXt [2], but **does not re-tune or ablate them specifically for 3D segmentation, nor validate them rigorously on more challenging datasets**. In our work, we explicitly repeat analogous ablations:
>
> - *Increasing model width* (from the ResNet-style `<64,128,256,512>` to the Swin Transformer-style `<96,192,384,768>`): this is reflected in rows 1 and 2 of Table 2.
> - *Enlarging the kernel size* of depthwise convolutions (from 3 to 7): this is reflected in rows 3 and 4 of Table 2.
>
> The results show that for 3D segmentation, these modifications bring only marginal performance gains while noticeably increasing computational cost. This is consistent with the observations reported in the MedNeXt paper, where
>
> - under the same kernel size, *scaling width* from S to L yields an average Dice improvement of less than 0.6%;
> - under the same number of channels, *enlarging the kernel* from 3 to 5 yields only about 0.1% average Dice improvement.
>
> These empirical findings motivate our focus on JLC, which preserves adjacency with a more favorable accuracy–efficiency trade-off than simply scaling depth-wise convolutions.

---

> > ### Author Response · Authors · 2025-11-22
> >
> > ### W4 & Q3: Possible data leakage in the generalization experiment comparing JLC and pruning methods
> >
> > Our previous external test set, MSD2019 Task 01, has substantial overlap (approximately 50%) with BraTS 2021: MSD2019 Task 01 contains 484 cases collected from BraTS 2016 and BraTS 2017 [5]. Among them, 244 TCIA cases from BraTS 2016 were explicitly removed in BraTS 2021 [6].
> >
> > To eliminate this overlap, we **restrict the external test set to these 244 TCIA cases** and re-run the evaluation. The updated results are as follows:
> >
> > | Method          | Avg    | ET     | TC     | WT     |
> > |-----------------|--------|--------|--------|--------|
> > | FULL w/o prune  | 79.97 | 77.35 | 79.39 | 83.17 |
> > | FULL w/ prune   | 62.00 | 63.66 | 64.47 | 57.86 |
> > | JL              | 79.34 | 76.69 | 78.37 | 82.95 |
> >
> > This confirms that our conclusions about the generalization behavior of JLC vs. pruning hold even after removing the overlapping cases and that the data leakage issue has been addressed.
> >
> > ### Q4: HD95 can help validate SKDT and contribute to rich texture details in feature learning
> >
> > Thank you very much for your suggestions on improving the SDKT experiments. Based on your suggestions, we **revise Table 4 and add changes to the HD95 metric**. In fact, a similar description of characterization optimization is also mentioned in [8].
> >
> > ### R1 References
> >
> > [1] Roy, Saikat, et al. "MedNeXt: Transformer-driven scaling of ConvNets for medical image segmentation." MICCAI, 2023.
> > [2] Liu Z, Mao H, Wu C Y, et al. "A ConvNet for the 2020s." CVPR, 2022.
> > [3] Isensee, Fabian, et al. "nnU-Net revisited: A call for rigorous validation in 3D medical image segmentation." MICCAI, 2024.
> > [4] Huang, Ziyan, et al. "Stu-Net: Scalable and transferable medical image segmentation models empowered by large-scale supervised pre-training." arXiv:2304.06716, 2023.
> > [5] Simpson A L, Antonelli M, Bakas S, et al. "A large annotated medical image dataset for the development and evaluation of segmentation algorithms." arXiv:1902.09063, 2019.
> > [6] Baid U, Ghodasara S, Mohan S, et al. "The RSNA-ASNR-MICCAI BraTS 2021 benchmark on brain tumor segmentation and radiogenomic classification." arXiv:2107.02314, 2021.
> > [7] Gatidis S, Hepp T, Früh M, et al. "A whole-body FDG-PET/CT dataset with manually annotated tumor lesions." Scientific Data, 2022.
> > [8] Qiu Z, Hu Y, Chen X, et al. "Rethinking dual-stream super-resolution semantic learning in medical image segmentation." TPAMI, 2023.

---

> > > ### Comment · Reviewer_kwQp · 2025-11-28
> > > **Comments 2 of 2**
> > >
> > > I wish to sincerely thank the authors for taking the data duplication issues seriously. This is an increasingly reoccurring issue with the usage of public datasets deriving data from the same source. I thank you for taking my comments seriously and putting in the leg work and also for the HD95. I intend to increase my score based on the rebuttal.

---

> > ### Comment · Reviewer_kwQp · 2025-11-28
> > **Comments 1 of 2**
> >
> > I thank the reviewers for providing the expanded experimentation. It would be remiss of me to not point out that the strongest variant of MedNeXt is infact it's L architecture and that ResEncL takes advantage of the nnUNet framework's autoconfig to maximize the model capacity and input patch size (based on the average data spacing) within a provided max VRAM size - something which in my opinion would not match other existing CNN baselines trivially. But I think I can still consider my somewhat convinced by the merits of the authors' comments in this direction.
> >
> > And of course, thank you for the improving the readability.

---

> ### Author Response · Authors · 2025-11-28
>
> Dear Reviewer kwQp,
>
> Thank you for your constructive comments and the increased score.
>
> MedNeXt is a highly representative CNN-based medical segmentation framework. We appreciate and incorporate this framework, which is why we did not conduct comparative experiments but instead demonstrated similar architectural performance in ablation experiments. Furthermore, in conjunction with the theme of efficient segmentation, we primarily employed the MedNeXt-S architecture in our experimental exploration, and further explored optimization strategies.
>
> Sincerely,
>
> Authors of Submission 3962

---

### Author Response · Authors · 2025-12-01
**Concluding Remarks for AC**

Dear Area Chair,

Thanks very much for your time and effort during this unusual review period. We submit this additional reply to briefly summarize how our rebuttal addresses the main concerns raised and to highlight the revisions made (which are also detailed in the **official comment**). Following the discussion, we are pleased that two reviewers responded positively to our rebuttal. Although the other two reviewers did not participate in the discussion, both indicated a willingness to increase their scores in their **original review**.

---
**Reviewer kwQp’s (original rating 4 / confidence 4, intends to increase)** main suggestions concerned the clarity of the PWA description, evidence that depth‑wise convolutions can harm adjacency, and the requirement of more baselines. In response, we replaced the original PWA analogy with a concise scientific explanation and added targeted experiments and ablations (e.g., with MedNeXt‑S) to provide direct empirical support.

Reviewer kwQp found these somewhat convincing and intended to **increase the score**.

---
**Reviewer syAZ’s (original rating 6 / confidence 4, had increased to 8)** main concerns were the core role of JLC, the mathematical details of JLC, and the additional overhead of SDKT. In response, we added ablations such as `DWC+PWA+SDKT` to confirm that JLC indeed serves as the foundation of our framework, highlighted the mathematical theory of JLC, and reported that SDKT incurs a low extra cost (~100 MB).

Reviewer syAZ found these clarifications thorough, and **raised the score to 8**.

---
**Reviewer L51F’s (original rating 6 / confidence 3)** main concerns were the need for K‑fold cross‑validation, the adequacy of the training epoch, and the rationale for our dataset choices. In response, we extended 4‑fold cross‑validation on four SOTA models while reporting mean/std/95%CI values, compared our total training iterations with nnU‑Net while showing loss curves, and explained that we deliberately focused on more challenging, heterogeneous, cross‑region datasets while adding comparative experiments on AMOS. We believe our rebuttal **has resolved most of Reviewer L51F’s concerns**

Reviewer L51F said: "I am open to discussing this further during the rebuttal and will be **happy to increase my score** if my concerns are addressed", but **did not join the discussion**.

---
**Reviewer PE39’s (original rating 2 / confidence 3)** comments mainly concerned: (i) clarification of code‑level details such as the fairness of the throughput calculation and the SDKT training scheme, (ii) interpretation of our results and the choice of primary metric, and (iii) additional experiments including multiple random seeds, more ablations, and scaling‑up.

We addressed (i) by referring to and further explaining the **anonymized repository** already included in the initial submission, (ii) by clarifying the primary metric and explaining anomalies in others, and (iii) by performing **all requested experiments**, even though some of them are not complemented in any of the 17 comparative works. Based on these clarifications and additions, we believe our rebuttal **has resolved most of Reviewer PE39’s concerns**.

Reviewer PE39 said: "I am **open to changing my score** based on the author's responses", but **did not join the discussion**.

---
We would like to sincerely thank all Area Chairs and reviewers once again for the significant time and effort devoted to the review process. We truly appreciate the careful and constructive feedback, which has helped improve the quality of our paper. We hope that the additional clarifications and analyses provided here will help support a fair and well-informed final decision.

---

### Meta-Review · Area_Chair_c7HU · 2026-01-09

**Summary:**

The reviewers acknowledged the strong empirical results and efficiency gains, but raised concerns about clarity and positioning of the main contributions, particularly the relative roles of JLC, PWA, and SDKT. Additional issues included the need for clearer explanations of PWA, fairer or more targeted baseline comparisons, stronger experimental rigor (e.g., cross-validation, training protocol justification), and clarification of domain generalization validity and evaluation metrics. One reviewer also questioned potential data overlap and code-level experimental details.

**Reviewer Concerns:**

The rebuttal substantially addressed many of the raised concerns by improving the clarity of the PWA description, adding targeted ablations and comparisons (including MedNeXt-S), extending cross-validation experiments, clarifying training scheme and throughput measurements, clarifying HD95 metric, and addressing the data overlap issue in the generalization experiments. The two reviewers who engaged with the rebuttal responded positively and explicitly acknowledged these improvements.

Some uncertainty remains regarding the precise positioning of JLC relative to PWA and SDKT, and the lack of direct comparison with the strongest large-scale baselines was discussed but not fully resolved. In addition, two reviewers did not participate in the discussion, so their final assessment can only be inferred indirectly.

**Reviewer Scores:**

The reviewers with initial scores of 6 and 4 explicitly stated that they would increase their scores following the rebuttal. The remaining two reviewers who did not engage in the discussion would likely keep their scores the same or increase them slightly. Taken together, this would plausibly lead to a virtual average score of around 5.5 or more, reducing the gap between reviews and moving toward consensus, although it is difficult to estimate exact outcomes without a direct reviewers discussion.

---

### Decision · Program_Chairs · 2026-01-26

Accept (Poster)